# Balancing reaction-diffusion network for cell polarization pattern with stability and asymmetry

Yixuan Chen[1,2,3†], Guoye Guan[1,4,5†‡, §], Lei-Han Tang[1,5*#], Chao Tang[3,4,6*#]

[1]South Bay Interdisciplinary Science Center, Songshan Lake Materials Laboratory, Dongguan, China; [2]Department of Physics, Sichuan University, Chengdu, China; [3]School of Physics, Peking University, Beijing, China; [4]Center for Quantitative Biology, Peking University, Beijing, China; [5]Department of Physics, Hong Kong Baptist University, Hong Kong, China; [6]Peking-Tsinghua Center for Life Sciences, Peking University, Beijing, China

**\*For correspondence:**
tangleihan@westlake.edu.cn (L-HT);
tangc@pku.edu.cn (CT)

[†]These authors contributed equally to this work

**Present address:** [‡]Department of Systems Biology, Harvard Medical School, Harvard Medical School, Boston, United States; [§]Department of Data Science, Dana-Farber Cancer Institute, Dana-Farber Cancer Institute, Boston, United States; [#]Center for Interdisciplinary Studies, Westlake University, Hangzhou, China

**Competing interest:** The authors declare that no competing interests exist.

## eLife Assessment

This manuscript makes **important** contributions to our understanding of cell polarization dynamics by demonstrating how compensatory regulatory and spatial mechanisms enhance the robustness of polarization patterns. By integrating a computational pipeline with comparisons to experimental data, the authors provide **convincing** evidence that stability and asymmetry in reaction-diffusion networks are crucial for polarization in *C. elegans* zygotes. Their findings offer novel insights into essential biological processes such as cell migration, division, and symmetry breaking. Future theoretical and experimental work could refine the model by addressing its acknowledged limitations.

**Abstract** Cell polarization is a critical process that separates molecular species into two distinct regions in prokaryotic and eukaryotic cells, guiding biological processes such as cell division and cell differentiation. Although several underlying antagonistic reaction-diffusion networks capable of setting up cell polarization have been identified experimentally and theoretically, our understanding of how to manipulate pattern stability and asymmetry remains incomplete, especially when only a subset of network components is known. Here, we present numerical results to show that the polarized pattern of an antagonistic 2-node network collapses into a homogeneous state when subjected to single-sided self-regulation, single-sided additional regulation, or unequal system parameters. However, polarity restoration can be achieved by combining two modifications with opposing effects. Additionally, spatially inhomogeneous parameters favoring respective domains stabilize their interface at designated locations. To connect our findings to cell polarity studies of the nematode *Caenorhabditis elegans* zygote, we reconstituted a 5-node network where a 4-node circuit with full mutual inhibitions between anterior and posterior is modified by a mutual activation in the anterior and an additional mutual inhibition between the anterior and posterior. Once again, a generic set of kinetic parameters moves the interface towards either the anterior or posterior end, yet a polarized pattern can be stabilized through tuning of one or more parameters coupled to intracellular or extracellular spatial cues. A user-friendly software, *PolarSim*, is constructed to facilitate the exploration of networks with alternative node numbers, parameter values, and regulatory pathways.

## Introduction

Cell polarization is a biophysical and biochemical process where a cell acquires spatial anisotropy by establishing directional gradients of molecular species across its membrane or in the cytosol (*Knoblich, 2001*; *Goodrich and Strutt, 2011*). Polarity establishment is an essential step in a wide range of biological phenomena, including embryonic development, wound healing, immune activity, and so forth (*Bilder et al., 2000*; *Etienne-Manneville and Hall, 2002*). During cytokinesis, a polarized cell can allocate its molecular contents unequally to its daughter cells, leading to asymmetric cell size and cell fate (*Macara, 2004*). At the multicellular level, a group of polarized cells can undergo collective movements and constitute stereotypical architectures (*Bilder et al., 2000*; *Etienne-Manneville and Hall, 2002*). Thus, loss or disorder of cell polarization could severely violate normal biological processes, for example, resulting in embryonic lethality and cancerous tumors (*Bilder et al., 2000*; *Kim et al., 2007*). To this day, cell polarization has been a long-term research focus in cell and developmental biology, where more and more efforts have been dedicated to uncovering both the regulatory pathways and design principles involved (*Doe and Bowerman, 2001*; *Tostevin and Howard, 2008*; *Chau et al., 2012*; *Koorman et al., 2016*).

Much of our knowledge on cell polarization is derived from the zygote of the nematode *Caenorhabditis elegans* (referred to as *C. elegans*), which has served as a prominent model organism for studying cell polarization (*Rose and Gönczy, 2014*; *Lang and Munro, 2017*). Following fertilization, the *C. elegans* zygote becomes polarized, with the sperm entry site (located at one pole of the ellipsoidal egg) designating the posterior of the embryo (*Goldstein and Hird, 1996*; *Motegi et al., 2011*). Driven by cell polarization, the embryo proceeds through four successive rounds of asymmetric cell divisions and produces four somatic founder cells sequentially (*Sulston et al., 1983*; *Hubatsch et al., 2019*).

The asymmetric division in the *C. elegans* zygote is governed by a protein family termed partitioning-defective protein (PAR) (*Kemphues et al., 1988*). The initial PAR family only consisted of the PAR-3/PAR-6/PKC-3 complex and PAR-1/PAR-2 complex, which are stably accumulated in the anterior and posterior domains on the membrane of the zygote before its division, respectively (*Cuenca et al., 2003*). During the establishment phase triggered by the sperm entry that defines the zygote's posterior, PAR-3/PAR-6/PKC-3, initially distributed across the entire cell membrane, are anteriorly directed and localized via the contraction of the cortical actomyosin marked by non-muscle myosin II (NMY-2), which allows the PAR-2 (along with the recruitment of PAR-1) to be enriched on the posterior membrane conversely (*Rose and Gönczy, 2014*; *Lang and Munro, 2017*). Subsequently, the maintenance phase follows as the cortical actomyosin contracts halfway across the zygote, with cortical NMY-2 being regulated downstream by PAR proteins and playing a minor role in maintaining the cell polarization pattern (*Goehring et al., 2011a*; *Beatty et al., 2013*; *Rose and Gönczy, 2014*). The cell polarization patterns, defined by the spatial concentration distributions of PAR proteins, are stably maintained to guide the assembly of the cytokinetic ring, which is vital for the first asymmetric cell division (*Goehring et al., 2011a*; *Rose and Gönczy, 2014*; *Jankele et al., 2021*).

A series of experiments (*incl.*, RNA interference, immunoprecipitations, fluorescence recovery after photobleaching, and time-lapse single-molecule imaging) further uncovered the mutual inhibition between PAR-3/PAR-6/PKC-3 and PAR-1/PAR-2 upon their association with the membrane and demonstrated its essential role in their robust spatial separation (*Cuenca et al., 2003*; *Motegi et al., 2011*). Soon, theoretical and numerical studies proved that mutual inhibition forms the backbone (hereafter referred to as an antagonistic 2-node network) of a polarized pattern (*Tostevin and Howard, 2008*; *Chau et al., 2012*). In the *C. elegans* zygote, the antagonism between these two protein groups directs the uneven distribution in downstream cell fate determinants (*e.g.*, PIE-1 and P granules), which is inherited by the two daughter cells during the following asymmetric cell division, leading to cell differentiation (*Schubert et al., 2000*; *Wang and Seydoux, 2013*).

In recent years, more proteins have been identified to significantly interact with the antagonistic 2-node network during the maintenance phase of *C. elegans* zygotic cell polarization, such as CDC-42, LGL-1, and CHIN-1 (*Beatty et al., 2010*; *Kumfer et al., 2010*; *Sailer et al., 2015*). The addition of these players increases the complexity of the cell polarization network tremendously, not only because of the explosion of the associated kinetic parameters, but also their role in related cellular processes such as cytoskeleton organization and localization at the poles that determines cell division dynamics (*Ajduk and Zernicka-Goetz, 2016*; *Lim et al., 2021*). Therefore, this growing complexity underscores

the importance of developing a more generalized understanding of the stability and asymmetry of cell polarization patterns, which will help to streamline experimental data analysis and interpretation. In a separate advancement, the theoretical knowledge acquired from nature is urgently needed for *de novo* construction of cells with designated characteristics, where the capability of cell polarization has been a design target for over a decade ago (*Tostevin and Howard, 2008*; *Chau et al., 2012*; *Lin et al., 2021*; *Watson et al., 2023*).

In this work, we focus on the generation of stable asymmetric patterns in both the widely-used 2-node network and a more realistic *C. elegans* 5-node network. Starting from a symmetric antagonistic network, a polarized pattern can be stabilized at any interface location between the two antagonistic domains, thereby manifesting translational symmetry. Unbalanced modification of kinetic parameters triggers movement of the interface in favor of one of the coexisting domains. Three types of such unbalanced modification are reported: single-sided self-regulation, single-sided additional regulation, and unequal system parameters. Nevertheless, the combination of two or more unbalanced modifications can recover pattern stability through fine-tuning of kinetic parameters. Alternatively, we show that the interface can also be stabilized at a designated location with a step-like spatial profile of one or more of the kinetic parameters, with values leading to opposing velocities when the interface is displaced. Intriguingly, such a program strategy is found to be employed in the *C. elegans* cell polarization network to maintain pattern stability along with considerable parameter robustness, while inducing pattern asymmetry by interface localization control.

## Results

### A computational pipeline for simulating cell polarization

To investigate both the simple network and the realistic network consisting of various node numbers and regulatory pathways (*Goehring et al., 2011b*; *Lang and Munro, 2017*), we propose a computational pipeline for numerical exploration of a given reaction-diffusion network's dynamics, specifically targeting the maintenance phase of stable cell polarization after its initial establishment (*Motegi et al., 2011*; *Goehring et al., 2011b*; *Seirin-Lee et al., 2020b*). Numerous biological experiments on different organisms have demonstrated that such a reaction-diffusion network typically comprises two groups of molecular species. Each group, once associated with a part of the cell membrane from cell cytosol, inhibits the association of the other group in the same region, thereby creating two types of distinct domains (*Knoblich, 2001*; *Doe and Bowerman, 2001*). At the interface between the two domains, the membrane association of proteins in either group is compromised due to the elevated level of antagonists. Nevertheless, one of the domains may expand at the expense of the other, leading to a finite interface velocity in general. For simplicity, three assumptions previously used in research are adopted to establish a reaction-diffusion model to describe the dynamics of each molecular species (denoted by $[X]$) during cell polarization:

1. The cellular space is reduced to a one-dimensional line of length $L = 0.5$, where $x = -0.25$ (anterior, where the concentration of molecular species accumulated on the cell membrane is denoted by $[A_m]$) and 0.25 (posterior, where the molecular species accumulated on the cell membrane is denoted by $[P_m]$) are its two poles (*Figure 1a*; *Gross et al., 2019*; *Seirin-Lee, 2020a*).
2. A molecular species $[X]$ can associate with the cell membrane from cell cytosol at a rate $F_{on}^{X}(x,t)$ and dissociate from the cell membrane into cell cytosol at a rate $F_{off}^{X}(x,t)$, where $t$ represents time. Both rates include leaky term and regulatory pathways affected by other molecular species (*Seirin-Lee and Shibata, 2015*; *Seirin-Lee et al., 2020b*).
3. Based on previous experimental measurements, it has been reported that the diffusion rate of molecular species involved in cell polarization (*i.e.*, PAR-2 and PAR-6) is two orders of magnitude higher in the cell cytosol compared to the cell membrane (*Goehring et al., 2011a*; *Blanchoud et al., 2015*; *Gross et al., 2019*; *Lim et al., 2021*). Consequently, we adopted $2.748 \times 10^{-5}$ and $1.472 \times 10^{-5}$ from previous research (nondimensionalized based on experimental measurements) (*Goehring et al., 2011a*; *Goehring et al., 2011b*; *Seirin-Lee et al., 2020b*), as diffusion coefficients of anterior and posterior molecular species, while the cytoplasmic molecules are regarded as well-mixed with infinite diffusion (*Kravtsova and Dawes, 2014*; *Gross et al., 2019*; *Morita and Seirin-Lee, 2021*). We further assume that these molecular species are sufficiently abundant in the pool of cell cytosol, where the concentration is conserved as constant $[X_c]$, although

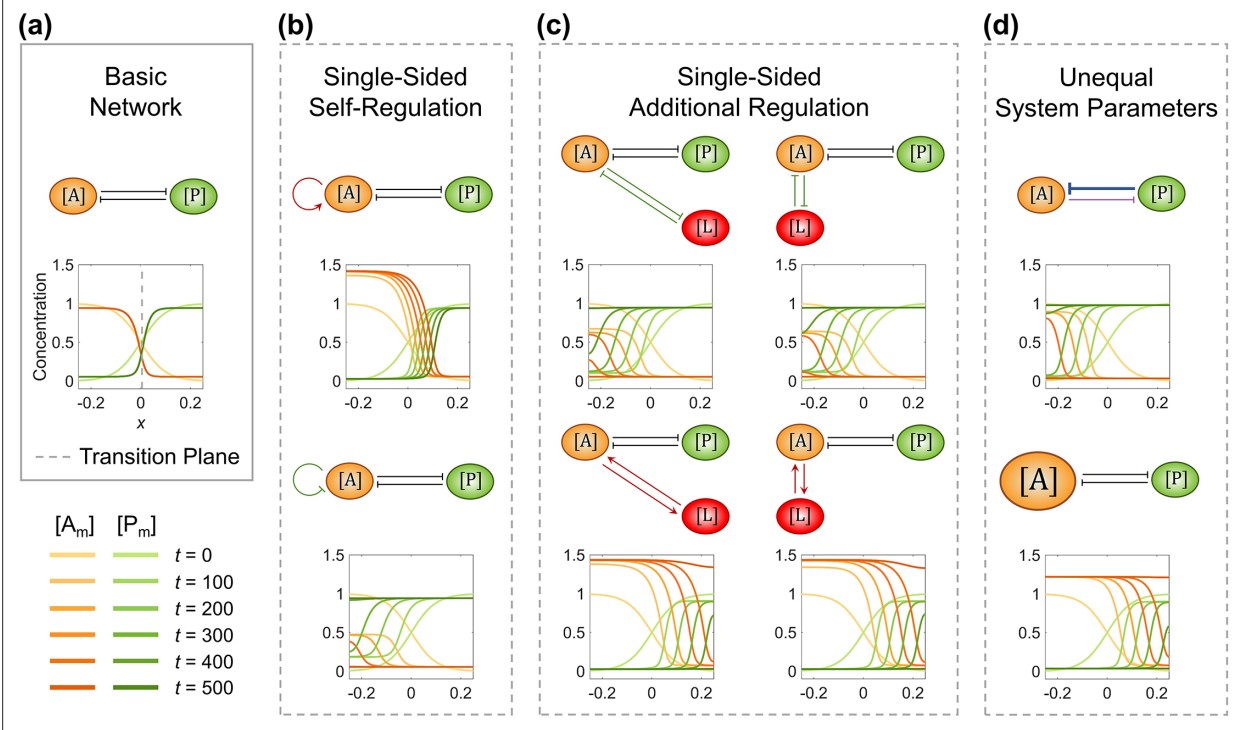

**Figure 1.** The single modification on the antagonistic 2-node network causes the collapse of cell polarization pattern. (**a**) Basic network with the transition plane (mid-plane) marked by gray dashed line. (**b**) Two subtypes of single-sided self-regulation. (**c**) Four subtypes of single-sided additional regulation. The corresponding spatial concentration distribution of $[L_m]$ at $t = 0, 100, 200, 300, 400,$ and $500$, and the subtle difference between left and right panels are detailed in *Figure 1—figure supplement 3*. (**d**) Two subtypes of unequal system parameters, exemplified by unequal inhibition intensity and unequal cytoplasmic concentration. For each network, the corresponding spatial concentration distribution of $[A_m]$ and $[P_m]$ at $t = 0, 100, 200, 300, 400,$ and $500$ are shown beneath with a color scheme listed in the bottom left corner. Note that within a network, normal arrows and blunt arrows symbolize activation and inhibition respectively.

The online version of this article includes the following video and figure supplement(s) for figure 1:

**Figure supplement 1.** The initial state at $t = 0$ (shown on the left) and stable state at $t = 500$ (shown in the middle) of the cell polarization pattern generated by the simple 2-node network and *C. elegans* 5-node network (shown on the right).

**Figure supplement 2.** The antagonistic 2-node network initiated with different initial spatial concentration distributions on the cell membrane.

**Figure supplement 3.** The antagonistic 2-node network added with four subtypes of single-sided additional regulation (corresponding to *Figure 1c*) causes the collapse of cell polarization pattern.

**Figure supplement 4.** With different initial spatial concentration distributions on the cell membrane, the single modification on the antagonistic 2-node network causes the collapse of cell polarization pattern.

**Figure supplement 5.** With parameters (*i.e.*, $\gamma$, $\alpha$, $k_1$, $k_2$, $q_1$, $q_2$, and $[X_c]$) assigned with various values concerning all nodes and regulations (detailed in *Appendix 1—table 2*, *Appendix 1—table 3*), the single modification on the simple 2-node network (top panel) and *C. elegans* 5-node network (bottom panel) causes the collapse of cell polarization pattern.

**Figure 1—video 1.** The spatial concentration distribution of each molecular species on the cell membrane over time generated by the antagonistic 2-node network and the ones with a single modification added, corresponding to *Figure 1*.

https://elifesciences.org/articles/96421/figures#fig1video1

the analysis below can be extended to the more general case (*Kravtsova and Dawes, 2014*; *Goehring et al., 2011b*).

For the concentration of each molecular species $[X]$ on the cell membrane, $[X_m]$, the evolution equation is described by:

$$\frac{\partial [X_m](x,t)}{\partial t} = D_m^X \frac{\partial^2}{\partial x^2}[X_m] + F_{on}^X(x,t)[X_c] - F_{off}^X(x,t)[X_m] \tag{1}$$

where $D_m^X$ is the diffusion coefficient of molecular species $[X]$ on the cell membrane. The first term on the right side represents the diffusion across the cell membrane; the second term represents the association with the cell membrane from cell cytosol; the third term represents the dissociation from the cell membrane into cell cytosol. These regulated association and dissociation are given by the Hill equation (*Seirin-Lee and Shibata, 2015*; *Seirin-Lee, 2021*):

$$F_{on}^X(x,t) = \gamma_X + \sum_Y \frac{q_{Y2}^X [Y_m]^{n_{Yq}^X}}{1 + q_{Y1}^X [Y_m]^{n_{Yq}^X}} \tag{2}$$

$$F_{off}^X(x,t) = \alpha_X + \sum_Y \frac{k_{Y2}^X [Y_m]^{n_{Yk}^X}}{1 + k_{Y1}^X [Y_m]^{n_{Yk}^X}} \tag{3}$$

where $F_{on}^X$ consists of a basal on-rate (*i.e.*, association rate) $\gamma_X$ and recruitment by other membrane-bound molecular species ($Y \neq X$) and itself ($Y = X$, *i.e.*, self-activation), quantified by positive parameters $q_{Yi}^X$ ($i = 1, 2$); $F_{off}^X$ consists of a basal off-rate (*i.e.*, dissociation rate) $\alpha_X$ and exclusion by other molecular species ($Y \neq X$) and itself ($Y = X$, *i.e.*, self-inhibition), quantified by positive parameters $k_{Yi}^X$ ($i = 1, 2$); the Hill coefficients $n_{Yq}^X$ and $n_{Yk}^X$ are set to 2 as used before (*Seirin-Lee and Shibata, 2015*; *Seirin-Lee, 2021*). To simplify the model for numerical exploration with an affordable computational cost, we set the same responsive concentration $q_{Y1}^X = k_{Y1}^X = k_1$ for both activation and inhibition pathways; the inhibition intensities $k_{Y2}^X$ are set to 1 as a numerically normalized value. We also set self and mutual activation parameters $q_{Y2}^X = q_2$ to be the same for synergistic proteins in the same group. Finally, the basal association and dissociation rates are regarded the same so that these spatially independent effects can be neutralized, in other words, $\gamma_X = \alpha_X = \gamma$. Consequently, there are only three independent parameters governing this system: $\gamma$, $k_1$, and $q_2$. A detailed description of the parameters is listed in *Appendix 1—table 1*. This dimensionally reduced parameter configuration is sufficient to describe the temporal evolution of cell polarization pattern on the membrane under the well-mixed cytoplasmic protein concentration. All the simplified parameter value assignments above will be extensively explored by giving different values to different molecular species and pathways as well as setting a heterogeneous spatial distribution.

Simulations are performed by systematically scanning the dimensionless parameter set $\Omega(\gamma, k_1, q_2)$ on a three-dimensional (3D) grid $\gamma \in [0, 0.05]$ in steps $\Delta\gamma = 0.001$, $k_1 \in [0, 5]$ in steps $\Delta k_1 = 0.05$, and $q_2 \in [0, 0.05]$ in steps $\Delta q_2 = 0.001$. The derived nondimensionalized parameter attains values on the same order of magnitude as those observed in reality, confirming the fidelity of the proposed model in representing the real system (Appendix 1). When it comes to a statement of switch from pattern stability to instability, the three-parameter simplification will be removed to free the parameter value assignments (*i.e.*, uniform random variables $\gamma \in [0, 0.05]$, $\alpha \in [0, 0.05]$, $k_1 \in [0, 5]$, $k_2 \in [0, 5]$, $q_1 \in [0, 5]$, $q_2 \in [0, 0.05]$, and $[X_c] \in [0.05, 5]$ added for setting up a network with cell polarization pattern stability, searched by the Monte Carlo method) concerning all nodes and regulations, auxiliarily validating the conclusion (*Fishman, 1995*).

Focusing on the cell polarization pattern stability and asymmetry during the maintenance phase, we simplify and set the initial distribution of molecular species on the cell membrane, $[X_m](x, 0)$, which is polarized during the establishment phase through active actomyosin contractility and flow (*Munro et al., 2004*; *Kravtsova and Dawes, 2014*), using the sigmoid function:

$$\begin{cases} [X_m](x, 0) = 1 - \frac{1}{1 + e^{-20x}}, & \text{for anterior molecular species} \\ [X_m](x, 0) = \frac{1}{1 + e^{-20x}}, & \text{for posterior molecular species} \end{cases} \tag{4}$$

Then the set of partial differential *Equation 1* evolves for 500 steps with a time step of 1 (*Figure 1—figure supplement 1*). The solution at $t = 500$ is saved for further analysis if:

(i) All the molecular species still have a polarized pattern, as defined by $[X_m](-L/2, 500) > 0.5$ and $[X_m](L/2, 500) < 0.5$ for the anterior molecular species and $[X_m](-L/2, 500) < 0.5$ and $[X_m](L/2, 500) > 0.5$ for the posterior molecular species.

(ii) The cell polarization pattern is stable or nearly intact over time for each molecular species, as defined by

$$\left| \frac{[\mathrm{X_m}](x_0, 500) - [\mathrm{X_m}](x_0, 499)}{[\mathrm{X_m}](x_0, 499)} \right| < 10^{-4}, \quad \forall x_0 \in [-0.25, 0.25] \tag{5}$$

Implemented on *Matlab* 2022b (*The MathWork Inc, 2022*), such a computational pipeline can describe the spatiotemporal dynamics of any reaction-diffusion network design, for evaluating its capability of maintaining cell polarization (*Figure 1—figure supplement 1*). Meanwhile, we construct an automatic software, *PolarSim*, for users to explore networks with arbitrarily set node numbers, parameter values, and regulatory pathways (Appendix 2, *Source code 1*).

## An unbalanced network structure or parameter leads to the collapse of a polarized pattern

Previous experimental and theoretical discoveries have uncovered the mutual inhibition between two molecular species as a fundamental design capable of generating cell polarization (*Kemphues et al., 1988*; *Cuenca et al., 2003*; *Tostevin and Howard, 2008*; *Chau et al., 2012*). Thus, we utilize the completely symmetric antagonistic 2-node network to investigate the behavior of its cell polarization pattern when the network structure or parameter is modified. The two nodes (*i.e.*, molecular species) placed in the anterior and posterior are denoted by $[\mathrm{A}]$ and $[\mathrm{P}]$. As there is no activation here then $q_2$ is not considered, the computational pipeline above establishes a total of 122 viable parameter sets $(\gamma, k_1)$ (122 among 5,151 sets, ~2.37%) that can achieve stable cell polarization. We use $(\gamma = 0.05, k_1 = 0.05)$ as a representative to show how the corresponding cell polarization pattern behaves under alternative initial conditions and elementary modification (*Figure 1*, *Figure 1—figure supplement 2*).

Regarding the initial conditions, while uniform initial distributions maintain a homogeneous pattern (*Figure 1—figure supplement 2a, b*), a polarized initial distribution reliably maintains the cell polarization regardless of variations in the initial distribution, such as shifts in transition planes (*Figure 1—figure supplement 2c, d*), weak polarization (*Figure 1—figure supplement 2e, f*), or asymmetric curve shapes between the two molecular species (*Figure 1—figure supplement 2f, g*). In contrast, an initial distribution with only Gaussian noise produces a multipolar pattern, with the two molecular species still locally excluding each other (*Figure 1—figure supplement 2h*). Altogether, these findings demonstrate the necessity of additional mechanisms, such as the active actomyosin contractility and flow, for establishing initial cell polarization independently of the reaction-diffusion network during the establishment phase (*Cuenca et al., 2003*; *Gross et al., 2019*).

Regarding the elementary modification, a total of three types are exerted on the node $[\mathrm{A}]$:

i.  Single-sided self-regulation (*Figure 1b*): a self-activation ($q_{\mathrm{A2}}^{\mathrm{A}} = 0.012$) or self-inhibition ($k_{\mathrm{A2}}^{\mathrm{A}} = 0.1$) is added on the node $[\mathrm{A}]$.

ii.  Single-sided additional regulation (*Figure 1c*, *Figure 1—figure supplement 3*): a new node $[\mathrm{L}]$ is added in the anterior or posterior, with mutual activation ($q_{\mathrm{L2}}^{\mathrm{A}} = q_{\mathrm{A2}}^{\mathrm{L}} = 0.012$) or inhibition ($k_{\mathrm{L2}}^{\mathrm{A}} = k_{\mathrm{A2}}^{\mathrm{L}} = 0.025$) with $[\mathrm{A}]$. The location of the additional node $[\mathrm{L}]$ decides the initial polarized pattern represented by the sigmoid function, where an anterior node starts with a high concentration in the anterior and a low concentration in the posterior, and vice versa.

iii.  Unequal system parameters (*Figure 1d*): on one hand, the inhibition intensity $k_2$ from $[\mathrm{P}]$ to $[\mathrm{A}]$ is increased from 1 to 1.6; on the other hand, the cytoplasmic concentration of $[\mathrm{A}]$ is increased from 1 to 1.25. The other system parameters (*e.g.*, basal on-rate $\gamma$ and basal off-rate $\alpha$) will be explored independently in the next section.

In comparison with the stable cell polarization pattern generated by the basic network, the spatial concentration distribution dynamics of eight modified conditions are simulated as shown in *Figure 1* and *Figure 1—video 1*, where the domain of one molecular species keeps invading the domain of the other one, ending in a homogeneous distribution. Such pattern instability caused by unbalanced network modification still holds when the network is initially set up using different initial conditions and parameters (*i.e.*, $\gamma$, $\alpha$, $k_1$, $k_2$, and $[\mathrm{X_c}]$) assigned with various values concerning all nodes and regulations (*Figure 1—figure supplement 4*, *Figure 1—figure supplement 5* – 1st row, *Appendix 1—table 2*). This intuitively reveals that the antagonistic 2-node reaction-diffusion network is prone to become unstable with its interface keeps moving when the perturbation is introduced to the network structure or parameter.

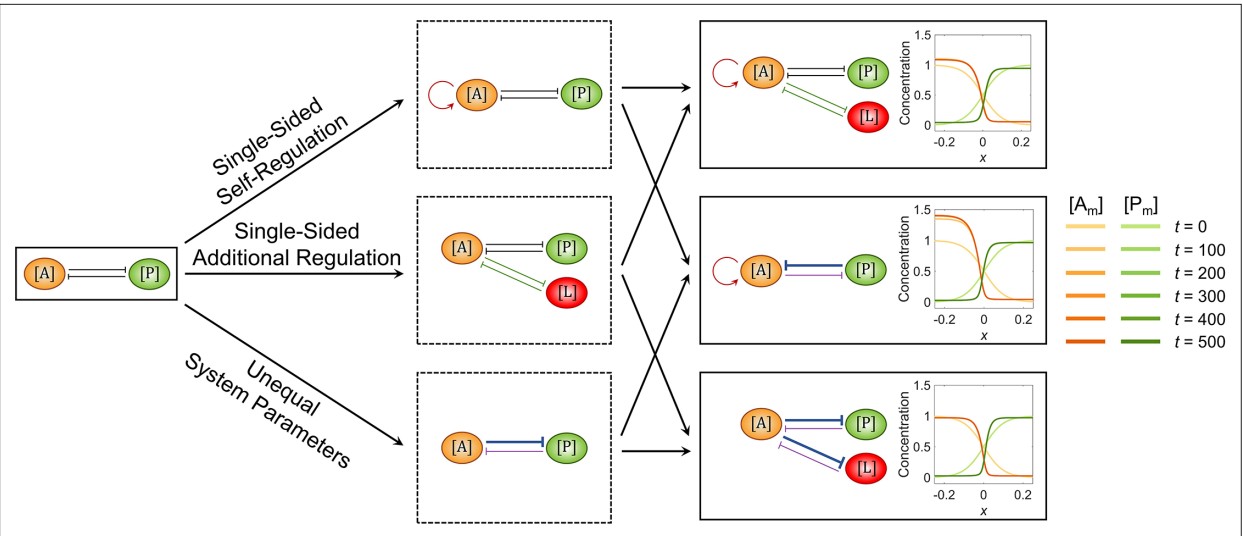

**Figure 2.** The combination of two opposite modifications recovers the cell polarization pattern stability. The basic network and the ones added with a single modification are shown in the 1st and 2nd columns, respectively; the three combinatorial networks composed of any two of the three single modifications are shown in the 3rd column. For each network, the corresponding spatial concentration distribution of $[A_m]$ and $[P_m]$ at $t = 0, 100, 200, 300, 400,$ and $500$ are shown beneath with a color scheme listed on the right. Here, the value assignments on the modifications in the 3rd column are as follows: $q_{A2}^A = 0.012$ and $k_{L2}^A \& k_{A2}^L = 0.01$ for 1st row, $q_{A2}^A = 0.012$ and $k_{P2}^A = 1.24$ for 2nd row, and $q_{A2}^A = 0.012$ and $k_{A2}^A \& k_{A2}^A = 2$ for 3rd row. Note that within a network, normal arrows and blunt arrows symbolize activation and inhibition respectively.

The online version of this article includes the following video for figure 2:

**Figure 2—video 1.** The spatial concentration distribution of each molecular species on the cell membrane over time generated by the antagonistic 2-node network and the ones with a single modification or two combinatorial modifications added, corresponding to *Figure 2*.
https://elifesciences.org/articles/96421/figures#fig2video1

## The combination of two modifications can recover the cell polarization pattern stability

Since a single modification of reaction-diffusion network structure or parameter is enough to break the cell polarization pattern, an appealing question just comes up: how can the network be designed to maintain pattern stability considering such modifications? This is crucial for cell polarization in reality where stability is essential for cell function and survival; without stability, the spatial information defined by the pattern is inaccurate for guiding downstream biological events such as cell division and cell differentiation (*Wang and Seydoux, 2013*; *Hubatsch et al., 2019*). The simplest idea for stability recovery is to combine two kinds of modifications with opposite trends, for example, adding self-activation and additional anteroposterior mutual inhibition on $[A]$ simultaneously. For the three types of modification, we arbitrarily select one subtype within each of them from *Figure 1b–d* so three combinatorial networks are constituted, all of whose interfaces turn out to be stabilized finally (*Figure 2*, *Figure 2—video 1*).

To more precisely elucidate how two opposite modifications coordinate the pattern stability together, we make use of the network with self-activation and additional inhibition on $[A]$ (shown in the top right corner of *Figure 2*), where the two corresponding intensities $q_{A2}^A$ and $k_{L2}^A \& k_{A2}^L$ compose a phase diagram that distinguishes the final state of the reaction-diffusion pattern. The pattern movement velocity is defined as follows:

$$v_M(t) = \frac{1}{N}\sum_X \frac{\int_{-\frac{L}{2}}^{\frac{L}{2}} \left| [X_m](x,t) - [X_m](x,t-1) \right| \mathrm{d}x}{L} \tag{6}$$

where $N$ is the total number of the molecular species. Here, we classify the final state by calculating the pattern movement velocity at the final time $t = 500$; the region with $v_M < 10^{-4}$ around the diagonal line is marked as the polarized state, while the regions upon and beneath it are homogeneous

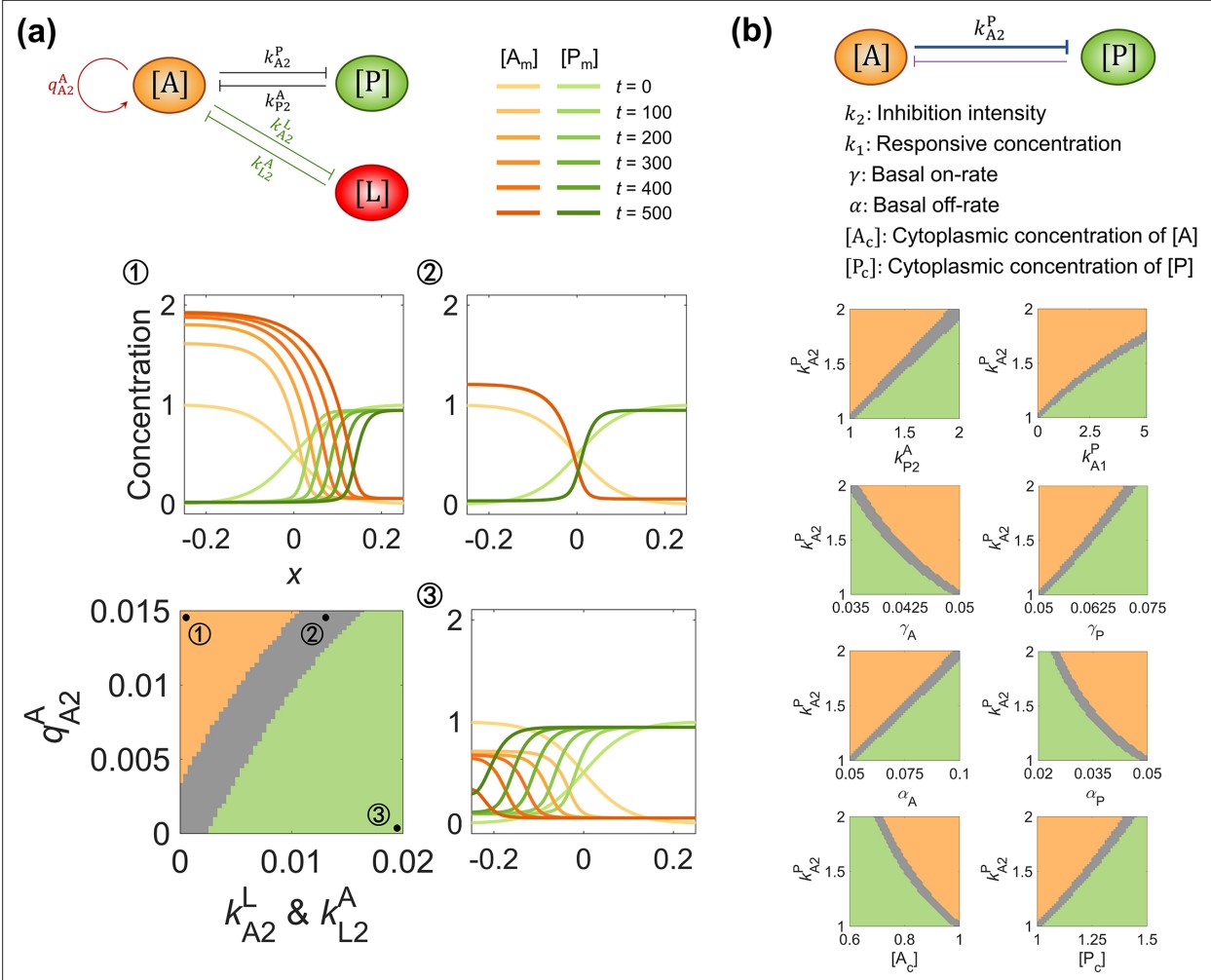

**Figure 3.** The balance between system parameters is needed for maintaining pattern stability. (**a**) The phase diagram between $q_{A2}^A$ and $k_{L2}^A$ & $k_{A2}^L$ in the network modified by self-activation (quantified by $q_{A2}^A$) and additional inhibition (quantified by $k_{L2}^A$ & $k_{A2}^L$) on $[A]$. The representative parameter assignment for each phase is marked with ① (*i.e.*, $q_{A2}^A = 0.015$ and $k_{L2}^A$ & $k_{A2}^L = 0$ with a homogeneous state dominated by $[A]$), ② (*i.e.*, $q_{A2}^A = 0.015$ and $k_{L2}^A$ & $k_{A2}^L = 0.0135$ with a stable polarized state), and ③ (*i.e.*, $q_{A2}^A = 0.02$ and $k_{L2}^A$ & $k_{A2}^L = 0$ with a homogeneous state dominated by $[P]$). The corresponding spatial concentration distribution of $[A_m]$ and $[P_m]$ at $t = 0, 100, 200, 300, 400,$ and $500$ are shown around the phase diagram with a color scheme listed on top. (**b**) The phase diagram between responsive concentration $k_1$, basal on-rate $\gamma$, basal off-rate $\alpha$, cytoplasmic concentration $[X_c]$, and inhibition intensity $k_2$. For each phase diagram in (**a**) (**b**), the final state dominated by $[A]$ or $[P]$ or stably polarized is colored in orange, green, and gray, respectively. Note that within a network, normal arrows and blunt arrows symbolize activation and inhibition respectively.

states dominated by $[A]$ and $[P]$ respectively (***Figure 3a***). Therefore, the triphase diagram and the exemplary patterns generated by the parameter assignments ①②③ shown in ***Figure 3a*** suggest the necessity of the balance between two opposite modifications for setting up a stable cell polarization pattern.

Apart from the unequal inhibition intensity and cytoplasmic concentration mentioned in ***Figure 1d***, we further ask if all the kinetic parameters with biophysical significance (*incl.*, inhibition intensity $k_2$, responsive concentration $k_1$, basal on-rate $\gamma$, basal off-rate $\alpha$, and cytoplasmic concentration $[X_c]$) also require a balance for achieving pattern stability. For this purpose, we generate the phase diagrams between each of them and the inhibition intensity $k_{A2}^P$. Fascinatingly, monotonic correlations exist between all those system parameters, suggesting that they can be tuned to maintain pattern stability (***Figure 3b***). In the gray region, the symmetric-broken parameters can be weighed against each other to realize the pattern movement velocity close to zero.

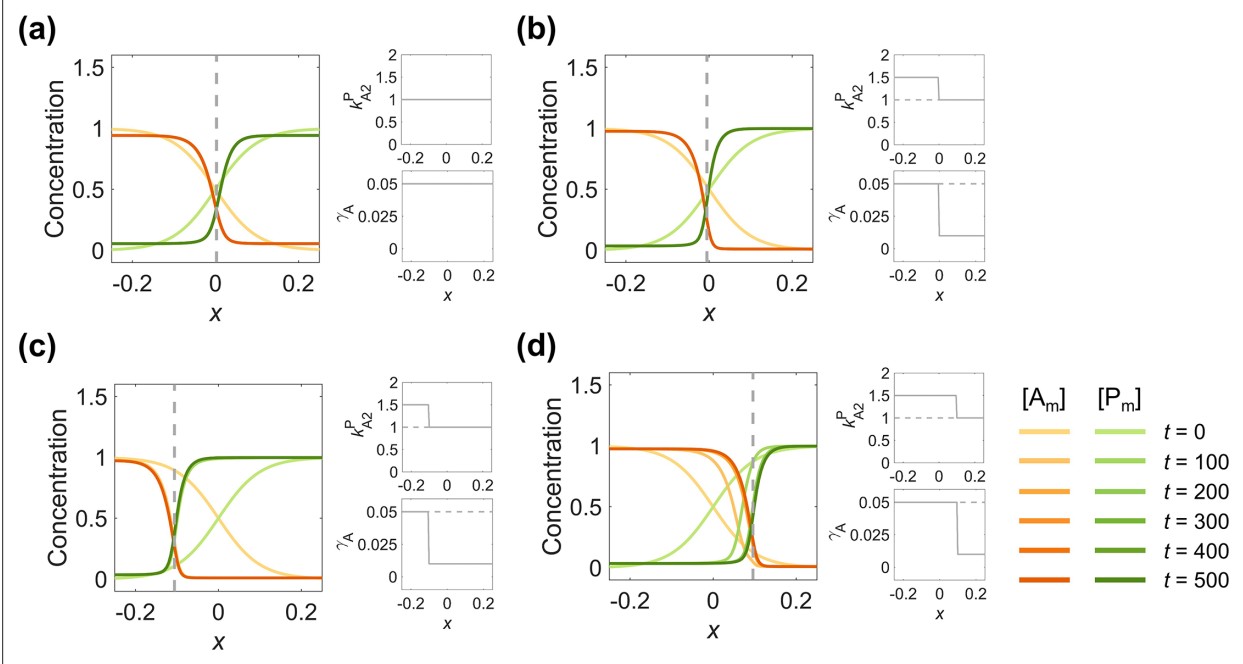

**Figure 4.** Adopting parameter sets corresponding to opposite interface velocities on two sides of the interface, the zero-velocity interface position of the polarity pattern is shiftable and regulable. (**a**) Spatially inhomogeneous parameters (2nd column) of a symmetric 2-node network generate a symmetric pattern (1st column), from the same simulation in *Figure 1a*. (**b**) Using the parameter combination with the posterior-shifting interface on the left and anterior-shifting interface on the right, a stable polarity pattern (1st column) can be remained by increasing $k_{A2}^{P}$ to 1.5 at $x < 0$ and decreasing $\gamma_A$ to 0.01 at (2nd column). (**c–d**) The stable interface position can be optionally adjusted by setting the change position of the step-up function (2nd column). (**c**) As in (**b**), but changing the step position to $x = -0.1$, the interface stabilizes around $x = -0.1$ (1st column). (**d**) As in (**b**), but changing the step position to $x = 0.1$, the interface stabilizes around $x = 0.1$ (1st column). For each parameter set, the corresponding spatial concentration distribution of $[A_m]$ and $[P_m]$ at $t = 0, 100, 200, 300, 400,$ and $500$ are shown beneath with a color scheme listed in the bottom right corner. Note that all the interface positions in (**a–d**) (1st column) are marked by gray dashed lines. Note that all the spatially homogeneous parameter distributions in (**a**) (2nd column) are marked by gray dashed lines in (**b–d**) (2nd column).

The online version of this article includes the following video for figure 4:

**Figure 4—video 1.** The spatial concentration distribution of each molecular species on the cell membrane over time generated by the antagonistic 2-node network with spatially inhomogeneous parameters, corresponding to *Figure 4*.

https://elifesciences.org/articles/96421/figures#fig4video1

## The velocity and position of the interface can be adjusted by setting up spatial cues

The required balance between parameters provides insight into controlling the pattern stability. However, in biological organisms, factors such as cellular actomyosin flow and intercellular signals could lead to changeable parameter values and molecular concentration in different parts of space (*Arata et al., 2010*; *Beatty et al., 2013*). How do cells maintain cell polarization pattern stability in the case of nonuniform initial conditions and parameter values, namely finding zero-velocity solutions at interfaces? Here, given a specific pattern, its position of the transition plane or interface $x_T(t)$ is defined by the mean position of all the molecular species where the absolute derivative value of the concentration curve reaches its maximum.

First, based on the symmetric 2-node network ($\gamma = 0.05$, $k_1 = 0.05$, *Figure 4a*, the same simulation as the one in *Figure 1a*), we alter its initial condition by shifting the interface position or exerting a uniform or Gaussian noise distribution without any positional information (*Figure 1—figure supplement 2*); it is found that the initial polarized pattern is essential for constricting only one single interface with predetermined position (*Figure 1—figure supplement 2c, d, e, f, g*), while the uniform or Gaussian noise distribution results in no or multiple interfaces (*Figure 1—figure supplement 2a, b, h*). This indicates the importance of additional mechanisms (*e.g.*, the cortical flow on the cell membrane that transports proteins related to cell polarity) establishing the initial cell polarization pattern before

the pattern enters the maintenance phase, when the network structure and parameters account for its stability in the long term (*Cuenca et al., 2003*; *Motegi et al., 2011*).

Next, to probe into how inhomogeneous biophysical parameter values can control the interface in concert, we employ the following adjustment on the parameter set used in *Figure 4a*:

(i) Increasing the inhibitory intensity of $[A]$ on $[P]$, $k_{A2}^P$, from 1 to 1.5, leads to the interface continuously shifting to the posterior, and $[A]$ finally dominates.

(ii) Decreasing the basal on-rate of $[A]$, $\gamma_A$, from 0.05 to 0.01, leads to the interface continuously shifting to the anterior, and $[P]$ finally dominates.

When these two systems are combined, the stable polarity pattern remains with (i) used in the region $x < 0$ while (ii) used in $x > 0$ (*Figure 4b*). By moving the step position of the parameter function to $x = -0.1$ (*Figure 4c*) and $x = 0.1$ (*Figure 4d*), the interface stabilizes around the step, which means the interface position is tunable (*Figure 4—video 1*). This indicates that by using parameter sets with values corresponding to opposite interface velocities on two sides of the interface, the interface velocity and profile of the polarity pattern can be precisely regulated. Such inhomogeneous spatial distribution of parameters imitates signaling perception, which provides a strategy for more robust control of the adjustable interface position in response to variable cues.

## Reconstruction of the molecular interaction network in *C. elegans* zygote and its design principle of structure and parameter trade-off

In the simple 2-node network, the cell polarization pattern stability could be broken by a single modification (*Figure 1*) and recovered by two combinatorial modifications (*Figures 2 and 3*), or with the zero-velocity interface position regulated by spatially inhomogeneous parameters (*Figure 4*). We further ask if such a fundamental rule is employed in the cell polarization network programming in a real system. To this end, we focus on the zygote of the nematode *C. elegans*, which has been a popular model for cell polarization study from both experimental and theoretical perspectives for more than three decades (*Kemphues et al., 1988*; *Goehring et al., 2011b*; *Lang and Munro, 2017*; *Lim et al., 2021*). By conducting an exhaustive literature search (a total of 19 references), we summarize the molecular interaction network in *C. elegans* zygote that consists of five interacting molecular species: PAR-3/PAR-6/PKC-3 complex (*abbr.*, $[A]$) and CDC-42 protein (*abbr.*, $[C]$) accumulated in the anterior, and PAR-1/PAR-2 complex (*abbr.*, $[P]$), LGL-1 (*abbr.*, $[L]$) and CHIN-1 (*abbr.*, $[H]$) accumulated in the posterior (*Figure 5a*). The detailed description of the biochemical mechanism of each regulatory pathway as well as the corresponding supporting references are listed in *Supplementary file 1*. Based on the network obtained experimentally, the computational pipeline described in Section (A computational pipeline for simulating cell polarization) establishes a total of 602 viable parameter sets $\Omega\left(\gamma, k_1, q_2\right)$ (602 among 262,701 sets, ~0.229%) that can achieve stable cell polarization (*Figure 5d and e*).

To verify whether the computational pipeline is reliable enough to simulate the protein distribution dynamics of the *C. elegans* network *in vivo*, we further reproduce a series of perturbation experiments reported previously. It's worth noting that the reconstructed *C. elegans* 5-node network appears to be an integration of an antagonistic 2-node network doubled into an antagonistic 4-node network (*Figure 5b*) with full connections, and the $[A]$~$[C]$ mutual activation (*Figure 5c*) and $[A]$~$[L]$ mutual inhibition (*Figure 5d*). As the symmetric 2-node network has been revealed to be a fundamental structure capable of stable cell polarization (*Tostevin and Howard, 2008*; *Goehring et al., 2011b*; *Chau et al., 2012*), hereafter we focus on how the added interactions involved with $[C]$ (the protein accumulated in the anterior and with an additional mutual activation with $[A]$ in the anterior) and $[L]$ (the protein accumulated in the posterior and with an additional mutual inhibition with $[A]$ in the anterior) impact the pattern stability concordantly (*Figure 5—figure supplement 1*).

Taking $[L]$ as the first example for its current absence in theoretical model analysis (*Seirin-Lee et al., 2020b*; *Lim et al., 2021*), a series of experimental groups in different conditions were conducted before by knocking down or overexpressing $[L]$, followed by a measurement of the lethality in embryo individuals (defined by death rate, a consequence of polarity loss in the zygote, which is essential for asymmetric cell division and cell differentiation) (*Beatty et al., 2010*; *Beatty et al., 2013*; *Rodriguez et al., 2017*; *Jankele et al., 2021*). Here, we utilize the pattern error in a mutant (*abbr.*, MT) embryo compared to the wild-type (*abbr.*, WT) one, $\text{Error} = \frac{1}{M}\sum_{i=1}^{M}\frac{\int_{-\frac{L}{2}}^{\frac{L}{2}}\left|[X_m]_{\text{WT}}(x,500) - [X_m]_{\text{MT}}(x,500)\right|dx}{L}$,

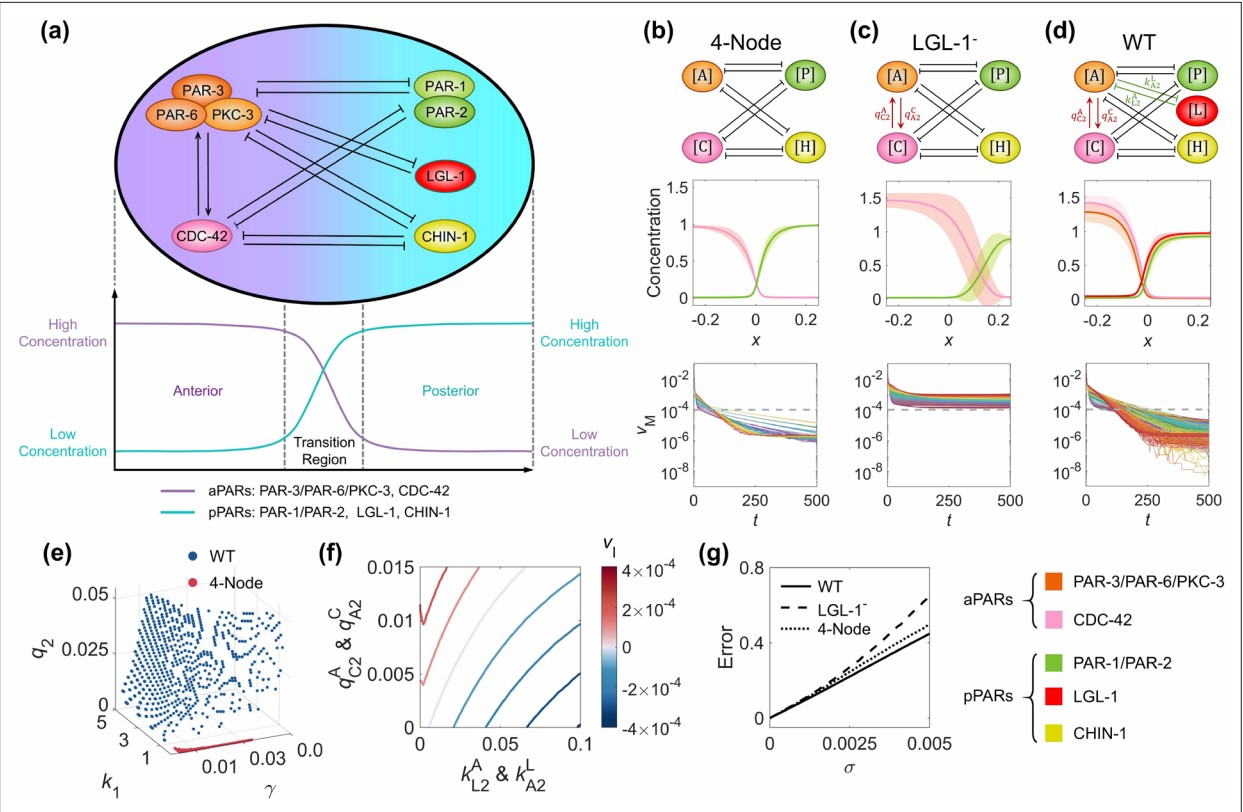

**Figure 5.** The molecular interaction network in *C. elegans* zygote and its natural advantages in terms of pattern stability, balanced network configuration, viable parameter sets, and parameter robustness. (**a**) The schematic diagram of the network composed of five molecular species, each of which has a polarized spatial concentration distribution on the cell membrane shown beneath. The left nodes PAR-3/PAR-6/PKC-3 (*i.e.*, $[\mathbf{A}]$) and CDC-42 (*i.e.*, $[\mathbf{C}]$) exhibit high concentration in the anterior pole and low concentration in the posterior pole, schemed by the purple line beneath; the right nodes PAR-1/PAR-2 (*i.e.*, $[\mathbf{P}]$), LGL-1 (*i.e.*, $[\mathbf{L}]$), and CHIN-1 (*i.e.*, $[\mathbf{H}]$) exhibit low concentration in the anterior pole and high concentration in the posterior pole, schemed by the cyan line beneath. Note that within the network, normal arrows and blunt arrows symbolize activation and inhibition respectively. (**b–d**) The structure of 4-Node, LGL-1⁻, and WT networks (1st row). The final spatial concentration distribution ($t = 500$) averaged over all established viable parameter sets for each molecular species (*i.e.*, 62 among 262,701 sets,~0.024%, for the 4-Node network; 62 among 5,151 sets,~1.2%, for the LGL-1⁻ network; 602 among 262,701 sets,~0.229%, for the WT network), shown by a solid line (2nd row). For each position, MEAN ±STD (*i.e.*, standard deviation) calculated with all viable parameter sets is shown by shadow. The pattern movement velocity ($v_\mathbf{M}$) over evolution time ($t$) (3rd row). For each subfigure in the 3rd row, each line with a unique color represents the simulation of a unique viable parameter set, and $v_\mathbf{M} = 10^{-4}$ is marked by a gray dashed line. For (**a–d**), the legend for the relationship between molecular species and corresponding color is placed in the bottom right corner. (**e**) The viable parameter sets of WT (blue points; 602 among 262,701 sets, ~0.229%) and 4-Node networks (red points; 62 among 5,151 sets, ~1.2%). (**f**) The interplay between $[\mathbf{A}]$~$[\mathbf{C}]$ mutual activation and $[\mathbf{A}]$~$[\mathbf{L}]$ mutual inhibition on the interface velocity. The contour map of the interface velocity $v_\mathbf{I}$ with different parameter combinations of $q_{C2}^A$ & $q_{A2}^C$ and $k_{L2}^A$ & $k_{A2}^L$ represents its moving trend. The darker the red is, the faster the interface is moving toward the posterior pole; the darker the blue is, the faster the interface is moving toward the anterior pole; the closer the color is to white, the stabler the interface is. (**g**) The averaged pattern error in a perturbed condition compared to the original 4-Node, LGL-1⁻, and WT networks. The Gaussian noise is simultaneously exerted on all the parameters $\gamma$, $\alpha$, $k_1$, $k_2$, $q_1$ and $q_2$ , where standard deviation $\sigma$ denotes the noise amplification. For a specific noise level, the pattern error is averaged over 1,000 independent simulations.

The online version of this article includes the following video and figure supplement(s) for figure 5:

**Figure supplement 1.** The transformation from the simple 2-node network to the *C. elegans* 5-node network.

**Figure supplement 2.** The *in silico* perturbation experiments on $[\mathbf{L}]$ based on the *C. elegans* modeling framework.

**Figure supplement 3.** The *in silico* perturbation experiments on $[\mathbf{C}]$ based on the *C. elegans* modeling framework.

**Figure supplement 4.** With different initial spatial concentration distributions on the cell membrane, the single modification on the *C. elegans* 5-node network causes the collapse of cell polarization pattern.

**Figure supplement 5.** The 32 possible additional feedback loops between LGL-1 (*abbr.* , $[\mathbf{L}]$) and PAR-3/PAR-6/PKC-3 (*abbr.*, $[\mathbf{A}]$) or PAR-1/PAR-2 (*abbr.*, $[\mathbf{P}]$).

**Figure 5—video 1.** The spatial concentration distribution of each molecular species on the cell membrane over time generated by the 4-Node, LGL-1⁻, and WT networks, corresponding to *Figure 5b–d*.
https://elifesciences.org/articles/96421/figures#fig5video1

to represent the level of polarity loss in simulation and lethality in experiment (measured in previous literature), where $M = 602$ is the number of viable parameter sets (602 among 262,701 sets, ~0.229%). The first experimental group is that the double depletion on $[P]$ and $[L]$ leads to much more lethality than single depletion on either $[P]$ or $[L]$, which is recapitulated by *Figure 5—figure supplement 2a, b* (cell polarization pattern characteristics in simulation: larger pattern error in double depletion on $[P]$ and $[L]$, compared to single depletion on either $[P]$ or $[L]$) (*Hoege et al., 2010*). The second experimental group is that the overexpression of $[L]$ lowers the lethality induced by single depletion on $[P]$ and such effect is weakened when $[H]$ is depleted as well, which is recapitulated by *Figure 5—figure supplement 2c,d* (cell polarization pattern characteristics in simulation: smaller pattern error in single depletion on $[P]$ supplemented by overexpression of $[L]$, compared to single depletion on $[P]$ only; less smaller pattern error in double depletion on $[P]$ and $[L]$ supplemented by overexpression of $[L]$, compared to single depletion on $[P]$ only) (*Beatty et al., 2010*; *Beatty et al., 2013*).

The second example of theoretical model analysis involves $[C]$. A series of experimental groups in different conditions were conducted before by knocking down or overexpressing $[C]$ or blocking the mutual activation between $[A]$ and $[C]$, followed by an observation of the cell polarity in embryo individuals. The first experimental group is that the single depletion on $[C]$ results in polarity defects and mislocalization of PAR-6 and PAR-2, which is recapitulated by *Figure 5—figure supplement 3a, b* (cell polarization pattern characteristics in simulation: less polarized distributions of $[A]$ and $[P]$, compared to WT) (*Gotta et al., 2001*; *Aceto et al., 2006*). The second experimental group is that the overexpression of $[C]$ increases the size of the anterior domain, which is recapitulated by *Figure 5—figure supplement 3a, c* (cell polarization pattern characteristics in simulation: posteriorly shifted interfaces, compared to WT) (*Aceto et al., 2006*). The third experimental group is that blocking the interaction between PAR-6 and CDC-42 results in polarity defects very similar to those associated with single depletion on CDC-42, which is recapitulated by *Figure 5—figure supplement 3a, d* (cell polarization pattern characteristics in simulation: less polarized distribution of $[A]$ in both conditions, compared to WT) (*Aceto et al., 2006*).

To sum up, apart from the cell polarization in a wild-type embryo, our modeling framework is further validated by reproducing two series of perturbation experiments, allowing further computational investigation of the network dynamics.

With the well-validated model of the *C. elegans* cell polarization network, here we study if it follows the balance design revealed by the exhaustive study on the antagonistic 2-node network. Interestingly, the central structure of the *C. elegans* network is shown to be a completely symmetric network composed of 2 nodes in the anterior (*i.e.*, $[A]$ and $[C]$) and posterior (*i.e.*, $[P]$ and $[H]$) respectively; this symmetric 4-node network is modified by a mutual activation in the anterior (*i.e.*, $[A]$~$[C]$) firstly and an additional mutual inhibition between the anterior and posterior (*i.e.*, $[A]$~$[L]$) subsequently, turning into an asymmetric 5-node network (*Figure 5b-d*, *Figure 5—figure supplement 1*). The abovementioned symmetric structure, the mutual activation, and the additional inhibition are in analogy with the basic 2-node network modified by self-activation and additional inhibition as shown in the top right corner of *Figure 2*, *Figure 2—video 1* and *Figure 3a*.

Here, we seek to compare the spatial concentration distribution and movement velocity of the patterns generated by the three successive network structures termed '4-Node' (the symmetric structure) (*Figure 5b*), 'LGL-1$^-$' (the symmetric structure added with $[A]$~$[C]$, that is the mutant network with $[A]$~$[L]$ eliminated from the wild-type network) (*Figure 5c*), and 'WT' (the symmetric structure added with both $[A]$~$[C]$ and $[A]$~$[L]$, *i.e.*, the wild-type network; *Figure 5d*). As expected, both simulations on the 4-Node and WT networks successfully pass the computational pipeline described in Section (A computational pipeline for simulating cell polarization) with 62 (62 among 5,151 sets, ~1.2%) and 602 (602 among 262,701 sets, ~0.229%) viable parameter sets respectively (*Figure 5e*), all of which stabilize into a cell polarization pattern with the spatial concentration distribution highly intact and the movement velocity of each parameter set continuously declining below $10^{-4}$ (*Figure 5b and d*, *Figure 5—video 1*). Nonetheless, the intermediate one (generated by depleting $[A]$~$[L]$ from the WT network) with $[A]$~$[C]$ but without $[A]$~$[L]$ fails (*Figure 5c*, *Figure 5—video 1*), exhibiting a dispersed concentration distribution and stubbornly high movement velocity for its pattern. Such pattern instability caused by unbalanced network modification still holds when the network is initially set up using different initial conditions and parameters (*i.e.*, $\gamma$, $\alpha$, $k_1$, $k_2$, $q_1$, $q_2$, and $[X_c]$) assigned with various values concerning all nodes and regulations (*Figure 1—figure supplement 5* – 2$^{nd}$ row, *Figure 5—figure*

supplement 4, *Appendix 1—table 3*), just like the simple 2-node network (*Figure 1—figure supplement 4*, *Figure 1—figure supplement 5* – 1st row, *Appendix 1—table 2*). Therefore, WT shows the largest variable parameter sets in the limited 3D grid and the strongest stability among the three network structures.

Remarkably, the balance between these two modifications is required to achieve stable cell polarization (*Figure 5f*). To evaluate how the interface moves, we quantify the transition plane or interface velocity by the rate of its position $x_T$ with respect to time. To avoid the influence of the initial state, the interface velocity is then calculated as the average velocity from $t = 300$ to $t = 500$.

$$v_I = \frac{x_T\,(t = 500) - x_T\,(t = 300)}{200} \tag{7}$$

Scanning the intensities of $[A]\sim[C]$ mutual activation (*i.e.*, $q_{A2}^C$ & $q_{C2}^A$) and $[A]\sim[L]$ mutual inhibition (*i.e.*, $k_{A2}^L$ & $k_{L2}^A$), a contour map is shown in consideration of the interface velocity averaged over all viable parameter sets (*i.e.*, $v_I$): the system can be stably polarized when they are in balance with its interface velocity close to zero, but the overshoot of either intensity leads to a homogeneous state in the end (*Figure 5f*). The redder the contour color is, the faster the interface moves posteriorly, with aPARs invading; the bluer the contour color is, the faster the interface moves anteriorly, with pPARs invading.

Next, we wonder if such combinatorial modifications may be an optimal choice selected during evolution. To this end, we regard the $[A]\sim[C]$ mutual activation as a primary modification that induces pattern asymmetry as proposed before (*Seirin-Lee et al., 2020b*; *Lim et al., 2021*), then the existence, as well as the form of the feedback loops between $[L]$ and a preexisting molecular species, is a supporting modification that consolidates stable cell polarization. Considering the symmetric structure of the LGL-1⁻ network, there is an identical role between $[A]$ and $[C]$ and between $[P]$ and $[H]$, so $[L]$ can be effectively connected to $[A]$ or $[P]$; meanwhile, there are three types of directional regulation between them: activation, inhibition, and none. Thus, in theory, there are 32 possible network structures with an additional regulation, in which $[L]$ must interact with one of the existing nodes (*Figure 5—figure supplement 5*). Strikingly, only the WT network with mutual inhibition between $[A]$ and $[L]$ passes the computational pipeline with viable parameter sets. Thus, without any parametric asymmetry concessions, the configuration of the *C. elegans* network in nature is well optimized among all other alternatives for maintaining cell polarization pattern stability.

As pattern stability means how fast the pattern moves over time, does the lack of pattern stability result in a more dispersed concentration distribution when the system parameters fluctuate in time? To test this hypothesis, for each viable parameter set, we exert Gaussian noise on all the original values of system parameters $\gamma$, $\alpha$, $k_1$, $k_2$, $q_1$, and $q_2$, and initiate 1000 independent simulations, where the noise amplification is represented by the standard deviation $\sigma$ of the Gaussian noise (*Guan et al., 2021*). To compare the variance of perturbed condition (*abbr.*, PT) to the original pattern (*abbr.*, OP, including, 4-Node, LGL-1⁻, and WT), the pattern error, averaged over all molecular species, all viable parameter sets, and all independent simulations, is defined as follows:

$$\text{Error} = \frac{1}{NMQ} \sum_{i=1}^{Q} \sum_{j=1}^{M} \sum_{X} \frac{\int_{-\frac{L}{2}}^{\frac{L}{2}} \left| [X_m]_{OP}\,(x, 500) - [X_m]_{PT}\,(x, 500) \right| \mathrm{d}x}{L} \tag{8}$$

where $N, M, Q$ represents the number of the molecular species types (*i.e.*, five for WT, four for 4-Node and LGL-1⁻), viable parameter sets (*i.e.*, 602 among 262,701, ~0.229%, for WT; 62 among 5,151, ~1.20%, for 4-Node; 62 among 262,701, ~0.024%, for LGL-1⁻), and independent simulations (*i.e.*, 1000). It turns out that the pattern error is always the smallest in the WT network and biggest in the LGL-1⁻ network, no matter how strong the noise is, indicating parameter robustness as the companion advantage of pattern stability (*Figure 5g*).

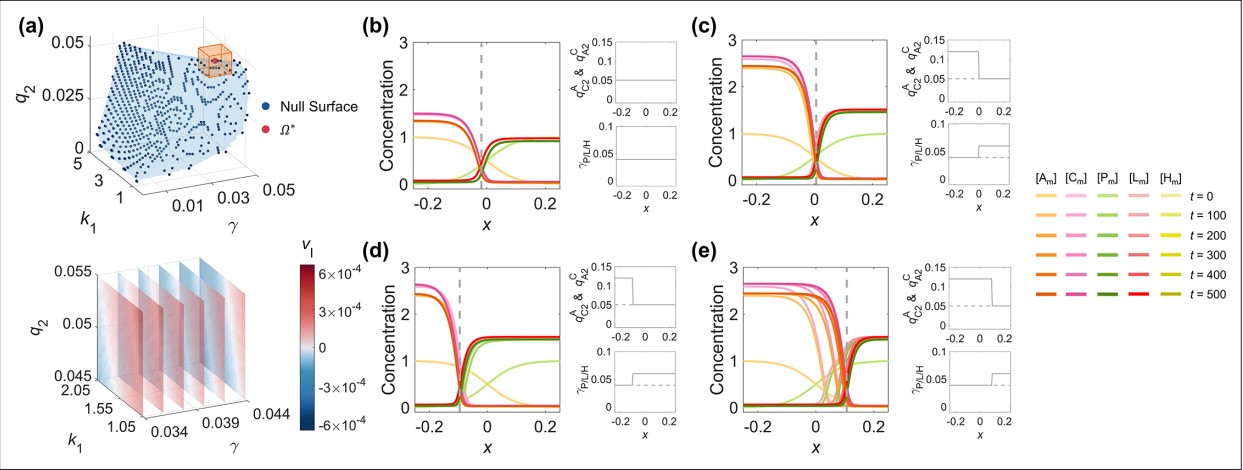

**Figure 6.** The control of the interface velocity and position by adjusting parameters in a multi-dimensional system. (**a**) The parameter space (1st row) and the linear relationship (2nd row) between interface velocity and parameters. The discrete viable parameter space of the WT network is fitted by a blue curved surface to represent its null surface with little or no pattern interface moving (1st row). The benchmark point $\Omega^*$ $(\gamma^* = 0.039, k_1^* = 1.55, q_2^* = 0.05)$ and its neighborhood are marked by an orange cube. Centering on the benchmark point $\Omega^*$, the relationship between the velocity interface and parameters in the region marked by the orange cube is shown by slice planes orthogonal to the $\gamma$-axis at the values 0.034, 0.036, 0.038, 0.04, 0.042, and 0.044 (2nd row). The darker the red is, the faster the interface is moving toward the posterior pole; the darker the blue is, the faster the interface is moving toward the anterior pole; the closer the color is to white, the stabler the interface is. (**b–e**) As in *Figure 4*, the control of the zero-velocity interface position of the polarity pattern by spatially inhomogeneous parameters corresponding to opposite interface velocities on two sides of the interface can be applied to the *C. elegans* 5-node network. (**b**) Using $\Omega^*$ as a representative, spatially homogeneous parameters (2nd column) generate a stable polarity pattern (1st column). (**c**) On top of (**b**), a stable polarity pattern (1st column) with its interface around $x = 0$ can be obtained by increasing $q_{C2}^A$ & $q_{A2}^C$ to 0.12 at $x < 0$ and increasing $\gamma_P, \gamma_L$, and $\gamma_H$ to 0.06 at $x > 0$ (2nd column). (**d**) As in (**c**) but changing the step position to $x = -0.1$ (2nd column), the interface stabilizes around $x = -0.1$ (1st column). (**e**) As in (**c**), but changing the step position to $x = 0.1$ (2nd column), the interface stabilizes around $x = 0.1$ (1st column). For each parameter set, the corresponding spatial concentration distribution of $[A_m], [C_m], [P_m], [L_m]$, and $[H_m]$ at $t = 0, 100, 200, 300, 400,$ and 500 are shown beneath with a color scheme listed on the right. Note that all the interface positions in (**b–e**) (1st column) are marked by gray dashed lines. Note that all the spatially homogeneous parameter distributions in (**b**) (2nd column) are marked by gray dashed lines in (**c–e**) (2nd column).

The online version of this article includes the following video for figure 6:

**Figure 6—video 1.** The spatial concentration distribution of each molecular species on the cell membrane over time generated by the *C. elegans* 5-node network with spatially inhomogeneous parameters, corresponding to *Figure 6b–e*.
https://elifesciences.org/articles/96421/figures#fig6video1

## A protocol to identify responsive parameters for interface positioning

For a cell polarization pattern to be stable, fine-tuning of the kinetic parameters is required to reach the vanishing interface velocity (numerically identified by *Equation 5*). Using the *C. elegans* 5-node network as an example, we outline a method to delineate iso-velocity surfaces $v_I(\Omega) = $ constant in the high-dimensional space of the parameter set $\Omega$. We show that the information gained can be used to quantify the role of individual molecular components in controlling the polarized pattern. Additionally, this knowledge enables us to design experiments to produce desired patterns.

The method of iso-velocity surface delineation is outlined as the following. In Section (Reconstruction of the molecular interaction network in *C. elegans* zygote and its design principle of structure and parameter trade-off), 602 (602 among 262,701, ~0.229%) groups of parameters capable of reaching stable polarity patterns are found by scanning through parameter space. When plotted in the 3D parameter space shown in *Figure 5e*, they span an iso-velocity surface (blue points in *Figure 5e*) at vanishing velocity, and in other words, their pattern interfaces are stabilized with little or no movement. To illustrate the local orientation of the surface, we selected 9,261 points (enumerated on a 3D grid $\gamma \in [0.034, 0.044]$ in steps $\Delta\gamma = 0.0005, k_1 \in [1.05, 2.05]$ in steps $\Delta k_1 = 0.05$, and $q_2 \in [0.045, 0.055]$ in steps $\Delta q_2 = 0.0005$) in the neighborhood of a benchmark point $\Omega^*$ $(\gamma^* = 0.039, k_1^* = 1.55, q_2^* = 0.05)$, as shown by the orange box in the top of *Figure 6a*. The least-squares fit of computed interface velocity to a linear function yields:

$$f(\gamma, k_1, q_2) = -0.0308 \times (\gamma - \gamma^*) - 6.08 \times 10^{-4} \times (k_1 - k_1^*) + 0.02 \times (q_2 - q_2^*) \tag{9}$$

with the coefficient of determination:

$$R^2 = 1 - \frac{\sum\limits_{i=1}^{9261} \left[ v_{\mathrm{I}}\left(\Omega_i\right) - f\left(\Omega_i\right) \right]^2}{\sum\limits_{i=1}^{9261} \left[ v_{\mathrm{I}}\left(\Omega_i\right) - \dfrac{1}{9261} \sum\limits_{j=1}^{9261} v_{\mathrm{I}}\left(\Omega_j\right) \right]^2} = 0.9918 \tag{10}$$

The coefficients in *Equation 9* give the ascending gradient of the interface velocity (*Figure 6a* – bottom).

Within the assumed relationships adopted for parameter reduction (see Section (A computational pipeline for simulating cell polarization)), *Equation 9* shows a strong dependence of the interface velocity on basal on/off-rates $\gamma$ and activation intensity $q_2$, but weak sensitivity to responsive concentration $k_1$ in the inspected parameter region. Interestingly, the increase in the basal on/off-rates $\gamma$ and self-activation rate $q_2$ have opposite effects on the interface velocity, which can be attributed to the asymmetric 5-node network structure. In a more realistic scenario, one may consider basal on/off-rates for different molecular species and their pairwise interactions separately and explore the relationships in a much larger parameter space. A formula similar to *Equation 9*, using a linear expansion around the parameter sets balanced for a zero-velocity interface, enables quantitative prediction of the interface velocity against individual or simultaneous changes of parameters, even when knowledge of the cell polarization network is incomplete. In particular, the parameter-dependent interface velocity picture potentially enables biologists to manipulate and even synthesize the polarity pattern by rational modulation of parameters, controlling the interface location for specific physiological functions, such as designating a desired cell volume partition ratio (*Hubatsch et al., 2019*; *Guan et al., 2021*).

Experimental studies have revealed that the interface localization in the *C. elegans* zygotic cell polarization pattern provides an accurate spatial cue to regulate the dynamics of their downstream molecules, like the protein LIN-5 and microtubules that control cell division and volume partition (*Schneider and Bowerman, 2003*; *Ajduk and Zernicka-Goetz, 2016*), while the acquired cell volume has been reported to have a chain effect in cell cycle, cell position, and other cell behaviors (*Fickentscher et al., 2018*; *Guan et al., 2021*; *Fickentscher et al., 2016*; *Tian et al., 2020*). As in the simple 2-node network, the precise control of the stable interface position can be realized by taking the parameters corresponding to the opposite direction of the interface velocity on both sides of the interface. Based on the parameter set $\Omega^*$ which gives out the stablest pattern (*Figure 6b*, *Figure 6—video 1*), two modifications are adopted to generate patterns with opposite interface velocities:

(i) Increasing the activation intensity between $[\mathrm{A}]$ and $[\mathrm{C}]$ ($q_{\mathrm{C2}}^{\mathrm{A}}$ & $q_{\mathrm{A2}}^{\mathrm{C}}$) to 0.12 results in the interface traveling backward, and $[\mathrm{A}]$ and $[\mathrm{C}]$ finally dominate the whole domain of the cell membrane.

(ii) Increasing the basal on-rate of posterior proteins ($\gamma_{\mathrm{P}}, \gamma_{\mathrm{L}}, \gamma_{\mathrm{H}}$) to 0.06 results in the interface traveling forward, and $[\mathrm{P}]$, $[\mathrm{L}]$ and $[\mathrm{H}]$ finally dominate the whole domain of the cell membrane.

Combining the two sets of parameters with its step switch located at $x = 0$ ((i) on the left and (ii) on the right), the stabilized polarity pattern interface settles around $x = 0$ (*Figure 6c*, *Figure 6—video 1*). The interface position turns tunable as the step switch moves to $x = -0.1$ (*Figure 6d*, *Figure 6—video 1*) and $x = 0.1$ (*Figure 6e*, *Figure 6—video 1*). One pre-known example in reality that matches this scheme is the asymmetric division of the P2 and P3 cells (the granddaughter and great-granddaughter cells of the *C. elegans* zygote P0), where the extracellular protein MES-1 and intracellular protein SRC-1 transduct signals from the EMS and E cells (the sister cell of P2 and its posterior daughter cell) and induce polarity reversal, possibly by affecting the local on-rate of PAR-2 (*Arata et al., 2010*; *Seirin-Lee, 2016*). Such a scheme depicts the introduction of intracellular or extracellular cues that break the spatial homogeneity of parameters and tune the stable localization of polarity pattern interface, serving as a theoretical basis for controlling oriented cell division with designated volume partition and fate differentiation (*Arata et al., 2010*; *Hubatsch et al., 2019*; *Schubert et al., 2000*).

# Discussion

Cell polarization is a fundamental issue in both prokaryotes and eukaryotes, playing crucial roles in diverse biological phenomena ranging from chemotaxis to embryogenesis (*Nance, 2014*; *Kondo et al., 2019*). The reaction-diffusion network responsible for cell polarization has been a long-term research focus for both experimentalists and theorists, who have identified many interactive functional molecules, discovered underlying design principles for networks, and even synthesized new systems *de novo* (*Tostevin and Howard, 2008*; *Chau et al., 2012*; *Koorman et al., 2016*; *Lang and Munro, 2017*; *Lin et al., 2021*; *Watson et al., 2023*). In this paper, we focus on a basic problem – how to control the cell polarization pattern stability over time and regulate the stabilized interface localization, from the perspective of a reaction-diffusion network. First, we established a computational pipeline to search the viable parameter sets in an *N*-node (molecular species) network that can achieve a stable cell polarization pattern. The simple antagonistic network with only two nodes was revealed to be unstable (*i.e.*, transit from a polarized distribution to a homogeneous distribution spontaneously) when any of the three unbalanced modifications (*i.e.*, single-sided self-regulation, single-sided additional regulation, and unequal system parameters) were introduced. To recover stable cell polarization, two strategies are proposed: (i) the combination of two unbalanced modifications with opposite effects; (ii) the combination of two spatially inhomogeneous parameter values, either of which can lead to opposite interface velocity. Additionally, the stable interface localization can be discretionarily regulated by the step-like parameter profile and contributes to the asymmetric geometry of the cell polarization pattern, potentially providing a spatial cue for significant physiological functions like unequal cell volume partition and consequent punctuated cell movement (*Fickentscher et al., 2016*; *Fickentscher et al., 2018*; *Tian et al., 2020*; *Guan et al., 2021*). Analogous conclusions are further identified in the experimentally summarized *C. elegans* network which obtains large parameter space and high robustness against parameter perturbations, supporting them as strategies indeed applied in real biological scenarios. Importantly, the linear relationship between the interface velocity and biophysical parameters serves as a useful tool to characterize and predict how the cell polarization pattern, especially where the interface is located or moved toward, responds to parameter changes; this not only helps understand how the regulatory molecular species and pathways, as well as the biophysical parameters learned from experiments, play their role, but also guides the design of artificial cell polarization systems with desired features. The joint study on a simple 2-node network and *C. elegans* 5-node network demonstrated that the cell polarization pattern with both stability and asymmetry can be explicitly realized by a combinatorial network and spatially inhomogeneous parameters, which is expected to facilitate the interpretation of natural systems and design of artificial systems.

Although we deciphered the stability and asymmetry of the pattern generated by the *C. elegans* cell polarization network, additional experimental details are required to achieve a more comprehensive understanding of this system, as outlined below.

1. Although the model in this paper has tried to incorporate as many molecular species as possible for the maintenance phase of cell polarization, the actual network may include yet unidentified molecular species. This also extends to interactions, as well as their intensity, between molecular species. For instance, the components of the molecular complex (*e.g.*, PAR-3/PAR-6/PKC-3), often treated as a single entity in theoretical studies (*Tostevin and Howard, 2008*; *Gross et al., 2019*; *Seirin-Lee et al., 2020b*; *Lim et al., 2021*), might not be ideally identical concerning their slightly different polarized patterns (*Wang et al., 2017*). The PAR-6 and PKC-3 also bind CDC-42, in addition to PAR-3, to mediate membrane association of CHIN-1 via phosphorylation, and such joint effect was previously described to be governed by a single node, CDC-42 (*Lim et al., 2021*). Besides, CDC-42 may inhibit CHIN-1 not through direct interaction but indirectly by recruiting the PAR-6/PKC-3 complex. Future experimental validation is needed to detail whether and how the molecular species described in *Supplementary file 1* (and beyond) interact with each other.
2. The large number of molecular species introduces substantial parameter complexity, with myriad possible value assignments. Simplifying the model is necessary to manage computational costs unless all parameters can be measured accurately *in vivo* (*Wang and You, 2020*).
3. Quantitative biological details, such as the spatial concentration distributions, diffusions, productions, and degradations of molecular species on the cell membrane versus in the cell cytosol (*Tostevin and Howard, 2008*; *Goehring et al., 2011b*; *Kravtsova and Dawes, 2014*; *Sailer et al., 2015*; *Seirin-Lee and Shibata, 2015*; *Seirin-Lee, 2016*; *Gross et al., 2019*; *Seirin-Lee*

*et al., 2020c*; *Lim et al., 2021*) and the precise mathematical forms of regulatory interactions (*e.g.*, Hill equations or product equations) (*Goehring et al., 2011b*; *Seirin-Lee et al., 2020b*; *Lim et al., 2021*) challenge model assumptions and hinder the unification of parameter value with unit between theory and experiment (*Tostevin and Howard, 2008*; *Kravtsova and Dawes, 2014*). Nonetheless, prior studies integrating theory and experiments have demonstrated that unveiling mathematical principles for regulatory networks does not strictly depend on unit unification; numerical network structure and parameter analysis can still effectively draw the mathematical principles for guiding the synthesis of artificial networks in reality, including the one for cell polarization (*Elowitz and Leibler, 2000*; *Gardner et al., 2000*; *Ma et al., 2009*; *Chau et al., 2012*).

4. Incorporating biological processes across temporal stages and spatial scales could offer a more holistic understanding of cell polarization. On one hand, early stages, such as the sperm entry and establishment phase, which involve active actomyosin contractility and flow driving PAR-3/PAR-6/PKC-3 to be localized in the anterior, can provide insights into how initial polarized concentration distributions are achieved (*Cuenca et al., 2003*; *Motegi et al., 2011*). On the other hand, molecular (*e.g.*, downstream localization of the myosin motor NMY-2 to the anterior cortex mediated by CDC-42 and PAR-2 *Goehring et al., 2011a*; *Rose and Gönczy, 2014*; *Geßele et al., 2020*) and cellular (asymmetric segregation of cell volume and molecular content, mediated by molecular species like MES-1 and SRC-1 *Arata et al., 2010*; *Seirin-Lee, 2016*) effects of cell polarization can further elucidate its various roles in embryogenesis.

Beyond the *C. elegans* zygote studied in this paper, cell polarization also exists in later stages of *C. elegans* embryogenesis as well as in other organisms, where diverse functional dynamics have been reported (*Knoblich, 2001*; *Rose and Gönczy, 2014*). Governed by the cell polarization, the *C. elegans* embryo actually proceeds through four rounds of asymmetric cell divisions to produce four somatic founder cells. Then, such a reaction-diffusion pattern will lose its sharp transition plane or interface as it is unscalable over cell size, leading to symmetric cell division at last (*Hubatsch et al., 2019*). Furthermore, as the reaction-diffusion network itself is not particularly constricted in the cellular scale, but also possibly in subcellular and multicellular scales, it would be interesting to investigate if the general principles proposed in this paper are also at work for the network programming and pattern modulation in other scales (*Chao et al., 2014*; *Guan et al., 2022*).

Nowadays, the *de novo* construction of artificial cells with designated new functions is emerging, demanding theoretical guidance about how to synthesize the molecular control circuits beneath (*Zhou et al., 2023*; *Zhu et al., 2023*). Like toggle switch, oscillation, and adaptation (*Gardner et al., 2000*; *Sun et al., 2022*; *Zhou et al., 2023*), the polarized concentration distribution of molecular species (accounting for cell polarization) is also another elementary behavior in a cell, and the corresponding circuits have been synthesized successfully over a decade ago (*Chau et al., 2012*). However, it remains unclear whether a molecular species' distribution pattern (*e.g.*, the transition plane or interface) can be altered while maintaining stability. Taking the *C. elegans* network as an example, our computational framework has been demonstrated to be capable of deciphering such an experimentally summarized reaction-diffusion network with pattern stability and asymmetry and has been packed as a user-friendly software, *PolarSim* (Appendix 2, *Source code 1*); thus we believe it can be used for not only understanding the natural molecular interaction networks reported before but also designing new molecular interaction networks in the polarized cells with more physiological functions, for example, with tunable transition plane or interface of molecular species to guide the allocation of downstream fate determinants and unequal cleavage during cytokinesis (*Fickentscher and Weiss, 2017*). In the future, our computational framework for cell polarization networks could be linked to the one about cytoskeleton activity, achieving a more comprehensive modeling description for cell division (*Pavin et al., 2012*; *Ma et al., 2014*).

## Acknowledgements

We thank Prof. Wei Wang, Dr. Daimin Li, Peng Xie, Shiyu Shen, and Qi Ding for their assistance in the numerical simulation, and Prof. Hongli Wang for his constructive advice on the project. We appreciate Zhengyang Han for his assistance in improving the paper materials. We are grateful to Dr. Siyu Li, Dr. Baoshuang Shang, Dr. Shipu Xu, and Yuqing Zhong for their help during Yixuan Chen and Guoye Guan's academic visit to Songshan Lake Materials Laboratory. Gratitude is extended to Prof. Zhongying Zhao, Dr. Dongying Xie, and Yiming Ma for their help during Guoye Guan's academic visit to

Hong Kong Baptist University. This work was supported by funding from the National Natural Science Foundation of China (12090053 and 32088101) and the Research Grants Council of the Hong Kong SAR (12303219). Computation was performed partly on the High-Performance Computing Platform at Peking University.

## Additional information

### Funding

| Funder | Grant reference number | Author |
|---|---|---|
| National Natural Science Foundation of China | 12090053 | Chao Tang |
| National Natural Science Foundation of China | 32088101 | Chao Tang |
| Research Grants Council of the Hong Kong SAR | 12303219 | Lei-Han Tang |

The funders had no role in study design, data collection and interpretation, or the decision to submit the work for publication.

### Author contributions

Yixuan Chen, Guoye Guan, Conceptualization, Data curation, Software, Formal analysis, Validation, Investigation, Visualization, Methodology, Writing – original draft; Lei-Han Tang, Chao Tang, Resources, Supervision, Funding acquisition, Validation, Investigation, Methodology, Project administration, Writing – review and editing

### Author ORCIDs

Yixuan Chen ⓘ https://orcid.org/0009-0004-3695-8752
Guoye Guan ⓘ https://orcid.org/0000-0003-4479-4722
Lei-Han Tang ⓘ http://orcid.org/0000-0002-2925-4691
Chao Tang ⓘ http://orcid.org/0000-0003-1474-3705

Joint Public Review: https://doi.org/10.7554/eLife.96421.3.sa1
Author response https://doi.org/10.7554/eLife.96421.3.sa2

## Additional files

### Supplementary files

Supplementary file 1. The literature summary on the cell polarization network in the *C. elegans* zygote.

MDAR checklist

Source code 1. The scripts and guidebook of the software *PolarSim* GUI.

### Data availability

The current manuscript is a theoretical and computational study, so no data has been generated for this manuscript. The computational pipeline for simulating cell polarization is packed as a user-friendly software, *PolarSim*, all associated source code is available on GitHub at https://github.com/YixuanChen0726/Cell-Polarization/tree/main/PolarSim (copy archived at *Chen, 2025*; also see Appendix 2 and *Source code 1*).

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

## Appendix 1

## Parameter nondimensionalization and order of magnitude consistency

The parameter nondimensionalization of the reaction-diffusion model is implemented according to a previous theoretical study (*Seirin-Lee et al., 2020b*). The dimensional evolution equation (corresponding to the nondimensional one in *Equation 1*) can be written as:

$$\frac{\partial \overline{[X_m]}}{\partial \bar{t}} = \overline{D_m^X} \frac{\partial^2}{\partial \bar{x}^2} \overline{[X_m]} + \left\{ \overline{\gamma_X} + \sum_Y \frac{\overline{q_{Y2}^X} \overline{[Y_m]}^2}{1 + \overline{q_{Y1}^X} \overline{[Y_m]}^2} \right\} \overline{[X_c]} - \left\{ \overline{\alpha_X} + \sum_Y \frac{\overline{k_{Y2}^X} \overline{[Y_m]}^2}{1 + \overline{k_{Y1}^X} \overline{[Y_m]}^2} \right\} \overline{[X_m]} \quad \text{(A1)}$$

where the dimensions (or units) of these original parameters are $\bar{t} \sim$ s, $\bar{x} \sim \mu$m, $\overline{[X_m]} \sim \overline{[X_c]} \sim \mu$m$^{-1}$, $\overline{D_X} \sim \mu$m$^2 \cdot$s$^{-1}$, $\overline{\gamma_X} \sim \overline{\alpha_X} \sim$ s$^{-1}$, $\overline{q_{Y1}^X} \sim \overline{k_{Y1}^X} \sim \mu$m$^2$, and $\overline{q_{Y2}^X} \sim \overline{k_{Y2}^X} \sim \mu$m$^2 \cdot$s$^{-1}$. Then we let $t = \frac{\bar{t}}{\tau}$, $x = \frac{\bar{x}}{x_0}$, $[X_m] = \frac{\overline{[X_m]}}{X_0}$, $[X_c] = \frac{\overline{[X_c]}}{X_0}$, and $[Y_m] = \frac{\overline{[Y_m]}}{Y_0}$, where $\tau$, $x_0$, $X_0$, and $Y_0$ are arbitrary constants with dimesions as s, $\mu$m, $\mu$m$^{-1}$, and $\mu$m$^{-1}$ correspondingly. In this paper, they adopt values from literature (*Seirin-Lee et al., 2020b*): $\tau = 2$ s and $x_0 = 142 \,\mu$m with $X_0 = Y_0 = 1 \,\mu$m$^{-1}$ (as a numerically normalized value). Finally, we derive the nondimensional evolution equation:

$$\frac{\partial [X_m]}{\partial t} = \frac{\overline{D_m^X}\tau}{x_0^2} \frac{\partial^2 [X_m]}{\partial x^2} + \left\{ \overline{\gamma_X}\tau + \sum_Y \frac{\overline{q_{Y2}^X} Y_0^2 \tau [Y_m]^2}{1 + \overline{q_{Y1}^X} Y_0^2 [Y_m]^2} \right\} [X_c] -$$
$$\left\{ \overline{\alpha_X}\tau + \sum_Y \frac{\overline{k_{Y2}^X} Y_0^2 \tau [Y_m]^2}{1 + \overline{k_{Y1}^X} Y_0^2 [Y_m]^2} \right\} [X_m]$$

$$\text{(A2)}$$

where the nondimensional parameters are expressed by the original ones as the following: $D_m^X = \frac{\overline{D_m^X}\tau}{x_0^2}$, $\gamma_X = \overline{\gamma_X}\tau$, $\alpha_X = \overline{\alpha_X}\tau$, $q_{Y1}^X = \overline{q_{Y1}^X} Y_0^2$, $q_{Y2}^X = \overline{q_{Y2}^X} Y_0^2 \tau$, $k_{Y1}^X = \overline{k_{Y1}^X} Y_0^2$, $k_{Y2}^X = \overline{k_{Y2}^X} Y_0^2 \tau$. *Equation A1* is equivalent to *Equation 1* in Section (A computational pipeline for simulating cell polarization).

Next, we reveal the consistency in order of magnitude between numerical and realistic values using four examples.

1. Evolution time: With a time step $\tau = 2$ s, the evolution time equals to $\tau = 500 \,\text{steps} \times 2 \,\text{s per step} = 16 \,\text{min} \,40 \,\text{s}$, in the same order of magnitude as the period of maintenance phase of *C. elegans* zygotic cell polarization, ~10 min, measured *in vivo* (*Blanchoud et al., 2015*).

2. Membrane diffusion coefficients: This was done in a previous theoretical study (*Seirin-Lee et al., 2020b*), based on experimental measurements (*Goehring et al., 2011a*; *Goehring et al., 2011b*; *Geßele et al., 2020*). Given the lengths of the major and minor semi-axis of the ellipsoidal *C. elegans* zygote are roughly $a = 27.0 \,\mu$m and $b = 14.8 \,\mu$m respectively, the perimeter of an ellipse after dimension reduction is roughly $x_0 = 2\pi b + 4(a - b) \approx 142 \,\mu$m. As the membrane diffusion rate of anterior PAR protein (exemplified by PAR-6) is $0.28 \,\mu$m$^2 \cdot$s$^{-1}$, while that of posterior PAR protein (exemplified by PAR-2) is $0.15 \,\mu$m$^2 \cdot$s$^{-1}$, their nondimensional values are determined as $D_m^X = \frac{\overline{D_m^X}\tau}{x_0^2} = \frac{0.28 \times 2}{142^2} \approx 2.748 \times 10^{-5}$ ($X \in [A, C]$) and $D_m^X = \frac{\overline{D_m^X}\tau}{x_0^2} = \frac{0.15 \times 2}{142^2} \approx 1.472 \times 10^{-5}$ ($X \in [P, L, H]$) respectively, which are used in this paper.

3. Basal off-rate: As the basal off-rate of anterior PAR protein (exemplified by PAR-6) is approximately $5.4 \times 10^{-3} \,$s$^{-1}$, while that of posterior PAR protein (exemplified by PAR-2) is approxiamtely $7.3 \times 10^{-3} \,$s$^{-1}$ (*Goehring et al., 2011a*; *Goehring et al., 2011b*), their nondimensional values are determined as $\alpha_X = \overline{\alpha_X}\tau = 5.4 \times 10^{-3} \times 2 = 0.0104$ ($X \in [A, C]$) and $\alpha_X = \overline{\alpha_X}\tau = 7.3 \times 10^{-3} \times 2 = 0.0146$ ($X \in [P, L, H]$) respectively, which match their scanned region $[0, 0.05]$ in this paper.

4. Inhibition intensity: As PAR-3/PAR-6/PKC-3 can be phosphorylated by PAR-2 with the rate of $0.02 \sim 0.8 \,\mu$m$^2 \cdot$s$^{-1}$ on the membrane, while PAR-1/PAR-2 can be released from the membrane by phosphorylation of PKC-3 with the rate of $0.06 \sim 1.6 \,\mu$m$^2 \cdot$s$^{-1}$ (*Geßele et al.,*

*2020*; *Goehring et al., 2011b*), their nondimensional values are determined as between $k^X_{Y2(MIN)} = \overline{k^X_{Y2(MIN)}} Y_0^2 \tau = 0.02 \times 1^2 \times 2 = 0.04$ and $k^X_{Y2(MAX)} = \overline{k^X_{Y2(MAX)}} Y_0^2 \tau = 1.6 \times 1^2 \times 2 = 3.2$, which match their scanned region $[0, 5]$ in this paper.

**Appendix 1—table 1.** The parameter description of the reaction-diffusion model for simulating a cell polarization network.

| Parameter | Biological implication | Defaulted nondimensional value in simulation |
|---|---|---|
| $x$ | Position along the anterior-posterior axis | [–0.25,0.25] in steps 0.005 |
| $t$ | Time | [0,500] in steps 1 (corresponding to 2 sec per step) (*Blanchoud et al., 2015*; *Seirin-Lee et al., 2020b*) |
| $[X]$ | Molecular species | / |
| $[X_m]$ | Spatial concentration distribution on the cell membrane | / |
| $[X_c]$ | Spatial concentration distribution in the cell cytosol | 1 |
| $D^X_m$ | Diffusion coefficient across the cell membrane | $2.748 \times 10^{-5}$ for anteriorly-located molecular species and $1.472 \times 10^{-5}$ for posteriorly-located molecular species (*Goehring et al., 2011a*; *Goehring et al., 2011b*; *Geßele et al., 2020*; *Seirin-Lee et al., 2020b*) |
| $\gamma_X$ | Basal on-rate (cell membrane association) | [0,0.05] in steps 0.001 |
| $\alpha_X$ | Basal off-rate (cell membrane dissociation) | [0,0.05] in steps 0.001 |
| $q^X_{Y1}$ | Responsive concentration in the activation pathway from $[Y]$ to $[X]$ | [0,5] in steps 0.05 |
| $k^X_{Y1}$ | Responsive concentration in the inhibition pathway from $[Y]$ to $[X]$ | [0,5] in steps 0.05 |
| $q^X_{Y2}$ | Activation intensity from $[Y]$ to $[X]$ | [0,0.05] in steps 0.001 |
| $k^X_{Y2}$ | Inhibition intensity from $[Y]$ to $[X]$ | 1 |
| $\Omega$ | A three-dimensional (3D) grid representing nondimensionalized parameter set | $(\gamma, k_1, q_2)$ |
| $n$ | Hill coefficient | 2 (*Seirin-Lee and Shibata, 2015*; *Seirin-Lee, 2021*) |
| $L$ | Cell/embryo length | 0.5 |
| $v_M$ | Pattern movement velocity | / |
| $x_T$ | Pattern transition plane or interface | / |
| $v_I$ | Interface velocity | / |
| $N$ | Number of molecular species types | / |
| $M$ | Number of viable parameter sets | / |
| $Q$ | Number of independent simulations | / |

**Appendix 1—table 2.** Viable parameter set searched by Monte Carlo method for the simple 2-node network.

For the 10 parameters in the simple 2-node network, the corresponding scanned ranges are $\gamma \in [0, 0.05]$, $\alpha \in [0, 0.05]$, $k_1 \in [0.05, 5]$, $k_2 \in [0.05, 5]$, and . Among 100,000 independent simulations, only 20 viable parameter sets pass the computational pipeline. One representative viable parameter set used to reproduce the conclusions in *Figure 1*; *Figure 1—figure supplement 5* -1st row is listed below.

| Parameter | Nondimensional Value in Simulation | Parameter | Nondimensional Value in Simulation |
|---|---|---|---|
| $[A_c]$ | 1.5361 | $[P_c]$ | 0.6662 |
| $\gamma_A$ | 0.0251 | $\alpha_A$ | 0.0437 |
| $\gamma_P$ | 0.0312 | $\alpha_P$ | 0.0245 |
| $k_{P1}^A$ | 0.1995 | $k_{P2}^A$ | 3.8471 |
| $k_{A1}^P$ | 1.3638 | $k_{A2}^P$ | 2.2571 |

**Appendix 1—table 3.** Viable parameter set searched by Monte Carlo method for the *C. elegans* 5-node network.

For the 39 parameters in the *C. elegans* 5-node network, the corresponding scanned ranges are $\gamma \in [0, 0.05]$, $\alpha \in [0, 0.05]$, $k_1 \in [0.05, 5]$, $k_2 \in [0.05, 5]$, $q_1 \in [0.05, 5]$, $q_2 \in [0, 0.05]$, and $[X_c] \in [0.05, 5]$. Among 100,000 independent simulations, only 6 viable parameter sets pass the computational pipeline. One representative viable parameter set used to reproduce the conclusions in *Figure 1*; *Figure 1—figure supplement 5* -2nd row is listed below.

| Parameter | Nondimensional Value in Simulation | Parameter | Nondimensional Value in Simulation |
|---|---|---|---|
| $[A_c]$ | 2.4420 | $[P_c]$ | 3.6739 |
| $[C_c]$ | 1.8330 | $[L_c]$ | 2.3793 |
| $[H_c]$ | 3.4683 | | |
| $\gamma_A$ | 0.0448 | $\alpha_A$ | 0.0310 |
| $\gamma_P$ | 0.0164 | $\alpha_P$ | 0.0334 |
| $\gamma_C$ | 0.0158 | $\alpha_C$ | 0.0071 |
| $\gamma_L$ | 0.0146 | $\alpha_L$ | 0.0221 |
| $\gamma_H$ | 0.0197 | $\alpha_H$ | 0.0410 |
| $q_{A1}^C$ | 4.1521 | $q_{A2}^C$ | 0.0495 |
| $q_{C1}^A$ | 1.3029 | $q_{C2}^A$ | 0.0201 |
| $k_{P1}^A$ | 3.7955 | $k_{P2}^A$ | 0.2009 |
| $k_{L1}^A$ | 2.3597 | $k_{L2}^A$ | 3.7357 |
| $k_{H1}^A$ | 1.9557 | $k_{H2}^A$ | 2.2441 |
| $k_{A1}^P$ | 3.7180 | $k_{A2}^P$ | 3.7342 |
| $k_{C1}^P$ | 4.5544 | $k_{C2}^P$ | 1.5792 |
| $k_{P1}^C$ | 0.5495 | $k_{P2}^C$ | 2.8838 |

*Appendix 1—table 3 Continued*

| Parameter | Nondimensional Value in Simulation | Parameter | Nondimensional Value in Simulation |
|---|---|---|---|
| $k_{H1}^{C}$ | 2.6518 | $k_{H2}^{C}$ | 1.2697 |
| $k_{A1}^{L}$ | 2.4436 | $k_{A2}^{L}$ | 4.8334 |
| $k_{A1}^{H}$ | 4.4505 | $k_{A2}^{H}$ | 0.3688 |
| $k_{C1}^{H}$ | 3.3712 | $k_{C2}^{H}$ | 2.0272 |

## Appendix 2

### Instructions for *PolarSim*

#### 2.1. Introduction

*PolarSim* is a graphical user interface (GUI) on *Matlab* 2022b (*The MathWork Inc, 2022*) for simulating the evolution of cell polarization patterns. Based on the reaction-diffusion model, the GUI allows users to compute the behaviors of cell polarization networks in different biological scenarios. All the simulations are tested with a 12th Gen Intel(R) Core(TM) i7-1260P CPU.

#### 2.2. Tutorials

- Download the folder "PolarSim" from https://github.com/YixuanChen0726/Cell-Polarization/tree/main/PolarSim.
- Open *MATLAB* under the 'PolarSim' folder path and execute script 'GUI.m'. Click 'Run' and then an interactive interface pops up (*Appendix 2—figure 1*).
- With the following parameters inputted, the GUI gives out a group of 'Pattern_*.mat' files containing pattern information.

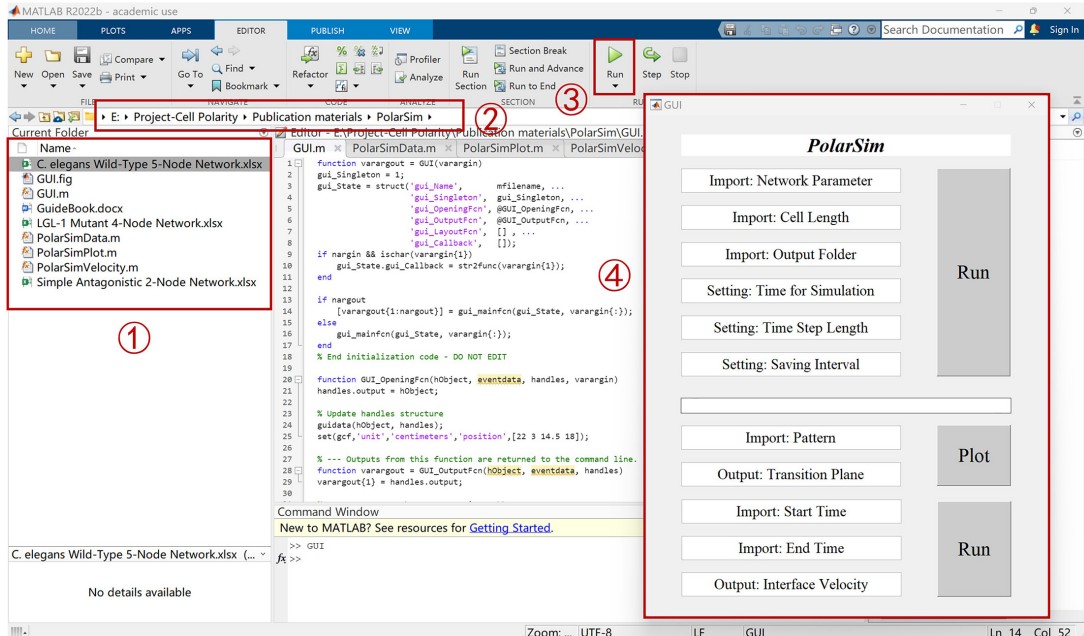

**Appendix 2—figure 1.** The instructions to open the *PolarSim* GUI. ① The files in the folder 'PolarSim'. ② Open *Matlab* under the path of the folder 'PolarSim' and double-click to open 'GUI.m'. ③ Click 'Run' to open the *PolarSim* GUI shown by ④.

**Appendix 2—table 1.** The instructions for the format of Network Parameter in an Excel table. In the 1st row and 1st column, 'Node N' represents the name of the node. The 2nd row explains the characteristic parameters of each node (*i.e.*, location, cytoplasmic concentration, basal on-rate, and basal off-rate as listed from left to right). The interaction parameters start from the 3rd row and 2nd column, where the $i$th row and $j$th column describe the activation/inhibition effect from Node $j$ to Node $i$. Note that $q_{Y2}^X$ and $k_{Y2}^X$ should be set to 0 when no activation or inhibition is exerted on X from Y respectively. 'Location' should be assigned with the string 'a' or 'p' while the description of the other parameters is detailed in *Appendix 1—table 1*.

| Node A | Node B | ... | Node N |
|---|---|---|---|
| Location (a or p)/ $[A_c]/\gamma_A/\alpha_A$ | Location (a or p)/ $[B_c]/\gamma_B/\alpha_B$ | ... | Location (a or p)/ $[N_c]/\gamma_N/\alpha_N$ |

*Appendix 2—table 1 Continued on next page*

*Appendix 2—table 1 Continued*

| | Node A | Node B | ... | Node N |
|---|---|---|---|---|
| Node A | $q_{A1}^A/q_{A2}^A/n_{Aq}^A/k_{A1}^A/k_{A2}^A/n_{Ak}^A$ | $q_{B1}^A/q_{B2}^A/n_{Bq}^A/k_{B1}^A/k_{B2}^A/n_{Bk}^A$ | ... | $q_{N1}^A/q_{N2}^A/n_{Nq}^A/k_{N1}^A/k_{N2}^A/n_{Nk}^A$ |
| Node B | $q_{A1}^B/q_{A2}^B/n_{Aq}^B/k_{A1}^B/k_{A2}^B/n_{Ak}^B$ | $q_{B1}^B/q_{B2}^B/n_{Bq}^B/k_{B1}^B/k_{B2}^B/n_{Bk}^B$ | ... | $q_{N1}^B/q_{N2}^B/n_{Nq}^B/k_{N1}^B/k_{N2}^B/n_{Nk}^B$ |
| ... | ... | ... | ... | ... |
| Node N | $q_{A1}^N/q_{A2}^N/n_{Aq}^N/k_{A1}^N/k_{A2}^N/n_{Ak}^N$ | $q_{B1}^N/q_{B2}^N/n_{Bq}^N/k_{B1}^N/k_{B2}^N/n_{Bk}^N$ | ... | $q_{N1}^N/q_{N2}^N/n_{Nq}^N/k_{N1}^N/k_{N2}^N/n_{Nk}^N$ |

**(a)**

| | A | B | C |
|---|---|---|---|
| 1 | | A | P |
| 2 | | a/1/2.748e-5/0.05/0.05 | p/1/1.472e-5/0.05/0.05 |
| 3 | A | 0.05/0/2/0.05/0/2 | 0.05/0/2/0.05/1/2 |
| 4 | P | 0.05/0/2/0.05/1/2 | 0.05/0/2/0.05/0/2 |

**(b)**

| | A | B | C | D | E | F |
|---|---|---|---|---|---|---|
| 1 | | A | P | C | L | H |
| 2 | | a/1/2.748e-5/0.039/0.039 | p/1/1.472e-5/0.039/0.039 | a/1/2.748e-5/0.039/0.039 | p/1/1.472e-5/0.039/0.039 | p/1/1.472e-5/0.039/0.039 |
| 3 | A | 1.55/0/2/1.55/0/2 | 1.55/0/2/1.55/1/2 | 1.55/0.05/2/1.55/0/2 | 1.55/0/2/1.55/0/2 | 1.55/0/2/1.55/1/2 |
| 4 | P | 1.55/0/2/1.55/1/2 | 1.55/0/2/1.55/0/2 | 1.55/0/2/1.55/1/2 | 1.55/0/2/1.55/0/2 | 1.55/0/2/1.55/0/2 |
| 5 | C | 1.55/0.05/2/1.55/0/2 | 1.55/0/2/1.55/1/2 | 1.55/0/2/1.55/0/2 | 1.55/0/2/1.55/0/2 | 1.55/0/2/1.55/1/2 |
| 6 | L | 1.55/0/2/1.55/1/2 | 1.55/0/2/1.55/0/2 | 1.55/0/2/1.55/0/2 | 1.55/0/2/1.55/0/2 | 1.55/0/2/1.55/0/2 |
| 7 | H | 1.55/0/2/1.55/1/2 | 1.55/0/2/1.55/0/2 | 1.55/0/2/1.55/1/2 | 1.55/0/2/1.55/0/2 | 1.55/0/2/1.55/0/2 |

**(c)**

| | A | B | C | D | E |
|---|---|---|---|---|---|
| 1 | | A | P | C | H |
| 2 | | a/1/2.748e-5/0.039/0.039 | p/1/1.472e-5/0.039/0.039 | a/1/2.748e-5/0.039/0.039 | p/1/1.472e-5/0.039/0.039 |
| 3 | A | 1.55/0/2/1.55/0/2 | 1.55/0/2/1.55/1/2 | 1.55/0.05/2/1.55/0/2 | 1.55/0/2/1.55/1/2 |
| 4 | P | 1.55/0/2/1.55/1/2 | 1.55/0/2/1.55/0/2 | 1.55/0/2/1.55/1/2 | 1.55/0/2/1.55/0/2 |
| 5 | C | 1.55/0.05/2/1.55/0/2 | 1.55/0/2/1.55/1/2 | 1.55/0/2/1.55/0/2 | 1.55/0/2/1.55/1/2 |
| 6 | H | 1.55/0/2/1.55/1/2 | 1.55/0/2/1.55/0/2 | 1.55/0/2/1.55/1/2 | 1.55/0/2/1.55/0/2 |

**Appendix 2—figure 2.** The examples of network parameters. (**a**) 'Simple Antagonistic 2-Node Network.xlsx' lists the parameters' in *Figure 1a*. (**b**) 'C. elegans Wild-Type 5-Node Network.xlsx' lists the parameters for the benchmark point $\Omega^*$ $\left(\gamma^* = 0.039, k_1^* = 1.55, q_2^* = 0.05\right)$ in *Figure 6b*. (**c**) 'LGL-1 Mutant 4-Node Network.xlsx' lists the parameters as in (**b**) but with the Node $\boxed{L}$ knocked out.

(i) Import: Network Parameter.
We give out three examples 'Simple Antagonistic 2_Node Network.xlsx', 'LGL-1 Mutant 4-Node Network.xlsx' and 'C. elegans Wild-Type 5-Node Network.xlsx' in the folder 'PolarSim', respectively representing the typical networks in this paper (*Appendix 2—figure 2*). The Excel table for parameter value assignments should follow the format in *Appendix 2—table 1*.

(ii) Import: Cell Length
We take '0.5' as an example. Any positive number is allowed in this box. The effects of cell length on cell polarization patterns are shown in *Appendix 2—figure 6*.

(iii) Import: Output Folder
Give a folder name for storing the output results (*e.g.*, 'Output 2-Node')

(iv) Setting: Time for Simulation
Simulation duration '500' is used in this paper. Any positive number is allowed in this box.

(v) Setting: Time Step Length
We take '1' as an example. Larger values are not recommended, in consideration of possible error accumulation, and one may try a smaller step length while the error tolerance is exceeded.

(vi) Setting: Saving Interval

It must be an integer multiple of the time step length. The time point will be saved in this designated interval and can be used for pattern plotting later.

- Click 'Run' in the interface, and then its status on the progress bar is shown (*Appendix 2—figure 3a*). A folder named 'Import: Output Folder' is generated in the current path to store the output 'Pattern_*.mat' data containing the molecular species or node's name, location,

and concentration distribution on the cell membrane, where '\*' denotes the *in silico* time corresponding to each file (*Appendix 2—figure 4a* - right).

- Import the pathway of the file outputted by *PolarSim* into the box 'Import: Pattern'.
- Click 'Plot' and then a figure pops up to show the cell polarization pattern while the position of the transition plane appears in the box 'Output: Transition Plane'.
- Import the pathway of two files outputted by *PolarSim* into the box 'Import: Start Time' and 'Import: End Time'.
- Click 'Run' and then the mean interface velocity between two input time points appears in the box 'Output: Interface Velocity'.

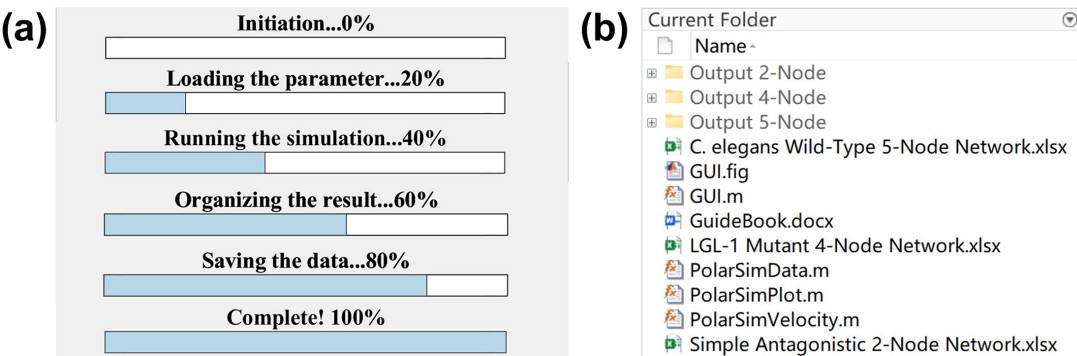

**Appendix 2—figure 3.** The progress bar, files, and subfolders of *PolarSim*. (**a**) The progress bar showing the running progress. (**b**) All files and output subfolders in the folder 'PolarSim'.

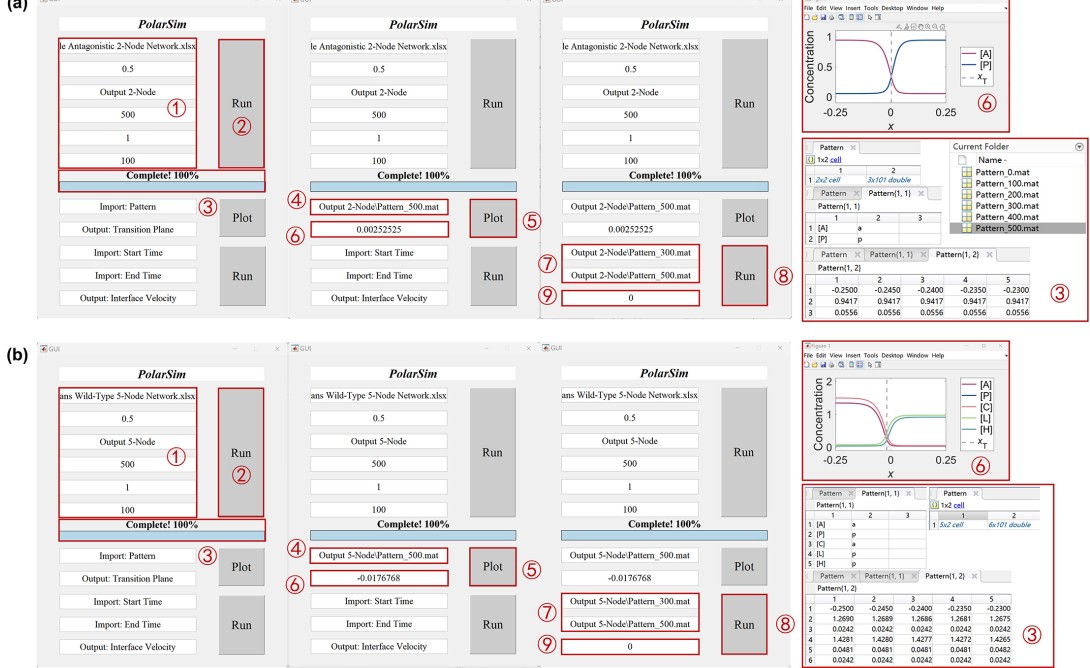

**Appendix 2—figure 4.** The results of Example 1 ('Simple Antagonistic 2-Node Network.xlsx') and Example 2 ('C. elegans Wild-Type 5-Node Network.xlsx') of *PolarSim*. (**a**) The flow chart for computing Example 1: ① input parameters; ② click 'Run'; ③ the simulation is completed with a progress bar shown and files saved in the folder 'Output 2-Node'; the file 'Pattern_500.mat' is used to show the data format on the right, where the first part stores the molecular species or nodes' names and locations while the second part stores their spatial concentration distributions on the cell membrane; ④ input the pathway of the outputted pattern file; ⑤ click 'Plot'; ⑥ the simulation is completed with a figure shown and the position of the transition plane given in the box 'Output: Transition Plane'; ⑦ input the pathway of the two outputted files 'Pattern_\*.mat' at different time points into *Appendix 2—figure 4 continued on next page*

*Appendix 2—figure 4 continued*
'Import: Start Time' and 'Import: End Time', where '*' denotes the start time and end time respectively; ⑧ click 'Run'; ⑨ the simulation is completed with the interface velocity given in the box 'Output: Interface Velocity'. (**b**) The same as **a** but for the 'C. elegans Wild-Type 5-Node Network.xlsx'.

## 2.3. Examples

**Example 1**: 'Simple Antagonistic 2-Node Network.xlsx' simulation as *Figure 1a*.

1. Input the following parameters: 'Simple Antagonistic 2-Node Network.xlsx' into 'Import: Network Parameter'; '0.5' into 'Import: Cell Length'; 'Output 2-Node' into 'Import: Output Folder'; '500' into 'Setting: Time for Simulation'; '1' into 'Setting: Time Step Length' and '100' into 'Setting: Saving Interval'. (*Appendix 2—figure 4a*– ①)
2. Click 'Run', and then six 'Pattern_*.mat' files at different time points are saved in the subfolder 'Output 2-Node' (*Appendix 2—figure 4a*– ②-③).
3. Input 'Output 2-Node\Pattern_500.mat' into the box 'Import: Pattern', and a figure at $t = 500$ appears. Then the position of the transition plane $x_T = 0.00252525$ is given in the box 'Output: Transition Plane' (*Appendix 2—figure 4a* – ④-⑥).
4. Input 'Output 2-Node\Pattern_300.mat' and 'Output 2-Node\Pattern_500.mat' into the box 'Import: Start Time' and 'Import: End Time' respectively. Then the interface velocity between and is given in the box 'Output: Interface Velocity' (*Appendix 2—figure 4a*– ⑦-⑨). The 2-node network approaches a stable polarized pattern with its interface velocity being 0.

**Example 2**: 'C. elegans Wild-Type 5-Node Network.xlsx' simulation as *Figure 6b*.

1. Input the following parameters: 'C. elegans Wild-Type 5-Node Network.xlsx' into 'Import: Network Parameter'; '0.5' into 'Import: Cell Length'; 'Output 5-Node' into 'Import: Output Folder'; '500' into 'Setting: Time for Simulation'; '1' into 'Setting: Time Step Length' and '100' into 'Setting: Saving Interval' (*Appendix 2—figure 4b*– ①).
2. Click 'Run'", and then six 'Pattern_*.mat' files at different time points are saved in the subfolder 'Output 5-Node' (*Appendix 2—figure 4b*– ②-③).

3. Input 'Output 5-Node\Pattern_500.mat' into the box 'Import: Pattern'", and a figure at $t = 500$ appears. Then the position of the transition plane $x_T = -0.0176768$ is given in the box 'Output: Transition Plane' (*Appendix 2—figure 4b*– ④-⑥).
4. Input 'Output 5-Node\Pattern_300.mat' and 'Output 5-Node\Pattern_500.mat' in the box 'Import: Start Time' and 'Import: End Time', respectively. Then the interface velocity between and is given in the box "Output: Interface Velocity" (*Appendix 2—figure 4b* – ⑦-⑨). The 5-node network approaches a stable polarized pattern with its interface velocity being 0.

**Example 3**: 'LGL-1 Mutant 4-Node Network.xlsx' simulation, originated from 'C. elegans Wild-Type 5-Node Network.xlsx' but with the Node $[L]$ knocked out.

1. Input the following parameters: 'LGL-1 Mutant 4-Node Network.xlsx' into 'Import: Network Parameter'; '0.5' into 'Import: Cell Length'; 'Output 4-Node' into 'Import: Output Folder'; '1000' into 'Setting: Time for Simulation'; '1' into 'Setting: Time Step Length' and '100' in 'Setting: Saving Interval' (*Appendix 2—figure 5a*).
2. Click 'Run', and then 11 'Pattern_*.mat' files at different time points are saved in the subfolder 'Output 4-Node'.

3. Input 'Output 4-Node\Pattern_500.mat' into the box 'Import: Pattern'", and then a figure of LGL-1 Mutant 4-Node Network at appears with the transition plane close to the posterior pole. Then the position of transition plane is given in the box 'Output: Transition Plane' (*Appendix 2—figure 5a*).

4. Input 'Output 4-Node\Pattern_300.mat' and 'Output 4-Node\Pattern_500.mat' in the box 'Import: Start Time' and 'Import: End Time' respectively. Then the interface velocity between $t = 300$ and $t = 500$ is given in the box 'Output: Interface Velocity' (**Appendix 2—figure 5a**). The interface of the 4-node network keeps moving toward the posterior with $v_I = 0.000429293$.

5. Input 'Output 4-Node\Pattern_1000.mat' in the box 'Import: Pattern', and then a figure at appears. The string 'Transition plane doesn't exist' appears in the box 'Output: Transition Plane' as the pattern reaches a homogeneous state dominated by and (**Appendix 2—figure 5b**).

6. Input 'Output 4-Node\Pattern_800.mat' and 'Output 4-Node\Pattern_1000.mat' in the box 'Import: Start Time' and 'Import: End Time', respectively. Then the interface velocity between and doesn't exist with the string 'It isn't a polarized pattern' appearing in the box 'Output: Interface Velocity' to hint (**Appendix 2—figure 5b**). Note that the interface velocity can't be calculated when either pattern at start time or end time is homogeneous.

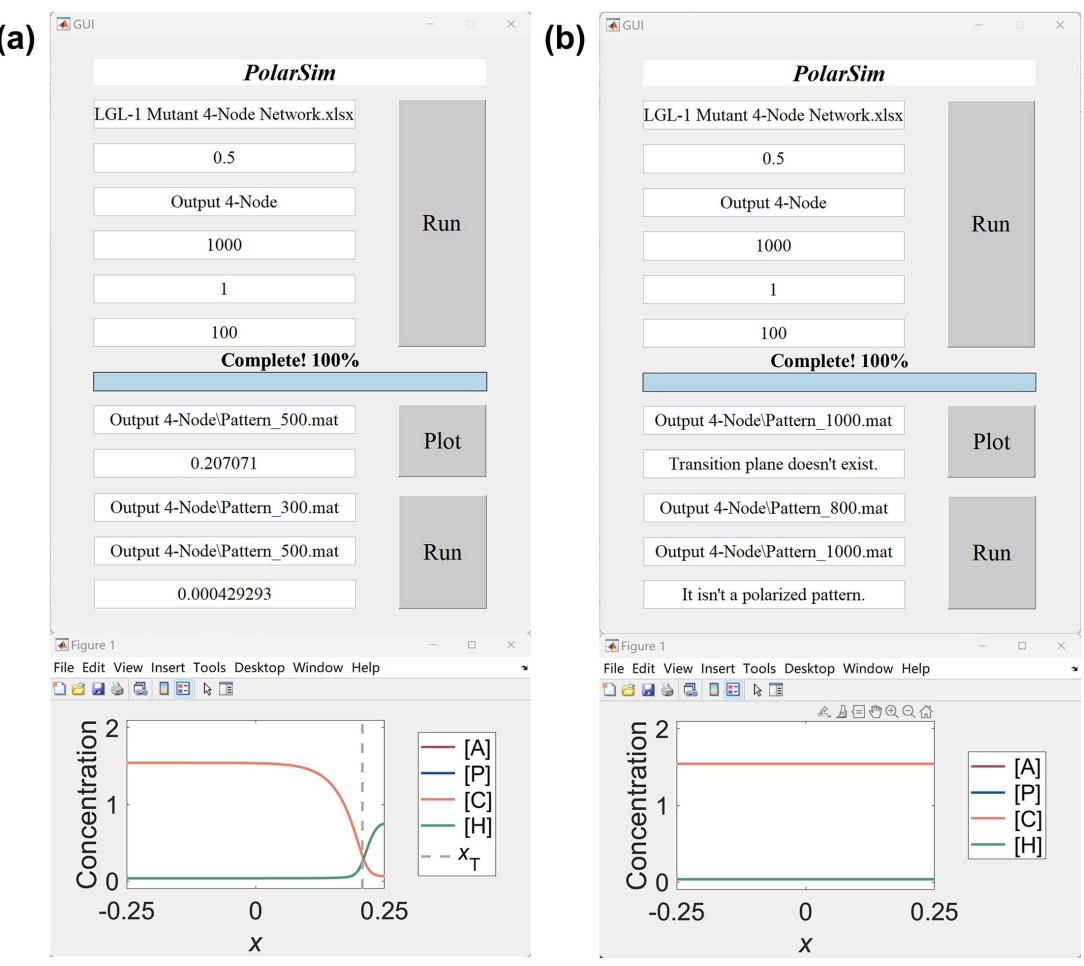

**Appendix 2—figure 5.** The results of Example 3 ('LGL-1 Mutant 4-Node Network.xlsx') of *PolarSim*. (**a**) The transition plane is shown at $t = 500$, with the value $x_T = 0.207071$. The interface velocity is calculated between $t = 300$ and $t = 500$, with the value $v_I = 0.000429293$ representing an unstable pattern (top). The figure is plotted at $t = 500$ (bottom). (**b**) The transition plane and the figure are shown at $t = 1000$ and the interface velocity is calculated between $t = 800$ and $t = 1000$. The pattern collapses to a homogeneous state with $[A]$ and $[C]$ invading the posterior domain at $t = 1000$, and thereby the transition plane doesn't exist and the interface can't be calculated.

## 2.4. Extensive application

Our *PolarSim* is extensively applicable to similar biological systems. Here, we take the cell size (length) as an exemplary research target to study how the concentration distribution on the cell membrane depends on it. Different cell lengths are applied to the Example 1 ('Simple Antagonistic

2-Node Network.xlsx') and Example 2 ('C. elegans Wild-Type 5-Node Network.xlsx') to see whether there is a cell size threshold limiting cell polarization as discovered before (*Hubatsch et al., 2019*). Patterns at *t* = 500 are plotted with the cell length ranging from 0.1 to 0.5 in steps 0.1 (*Appendix 2— figure 6*).

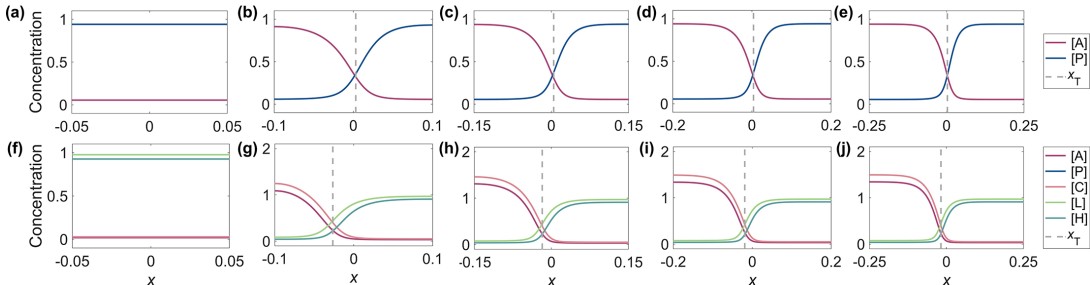

**Appendix 2—figure 6.** The effects of cell size (length) on the cell polarization pattern. (**a–e**) The pattern of 'Simple Antagonistic 2-Node Network.xlsx' at *t* = 500. From left to right, the cell lengths are 0.1, 0.2, 0.3, 0.4 and 0.5, respectively. (**f–j**) The same as (**a–e**) but for the 'C. elegans Wild-Type 5-Node Network.xlsx'.

The *PolarSim*-based simulations above indicate that the proper cell polarization demands a reasonable spatial scale, which to some extent gives an explicit constraint for the volume of a cell in reality when it needs to divide asymmetrically by reading out its cell polarization pattern interface (*Hubatsch et al., 2019*). It's worth noting that our prediction of cell polarization pattern collapse (with less and less distinguishable interface) over cell size decrease aligns with previous experimental and theoretical research (*Hubatsch et al., 2019*). In all, *PolarSim* provides a user-friendly tool for more applications on the studies in cell polarization.

## 2.5. Contact

All the scripts of the *PolarSim* GUI have been uploaded onto GitHub https://github.com/YixuanChen0726/Cell-Polarization/tree/main/PolarSim. If there is any question, please contact Yixuan Chen (yixuanchen@stu.pku.edu.cn) or Guoye Guan (guanguoye@gmail.com) anytime.

