## [Editor Report · eLife Assessment]

This manuscript makes **important** contributions to our understanding of cell polarization dynamics by demonstrating how compensatory regulatory and spatial mechanisms enhance the robustness of polarization patterns. By integrating a computational pipeline with comparisons to experimental data, the authors provide **convincing** evidence that stability and asymmetry in reaction-diffusion networks are crucial for polarization in *C. elegans* zygotes. Their findings offer novel insights into essential biological processes such as cell migration, division, and symmetry breaking. Future theoretical and experimental work could refine the model by addressing its acknowledged limitations.

---

## [Referee Report · Joint Public Review]

In this manuscript, the authors aim to evaluate the robustness of stable asymmetric polarization patterns by analyzing both a minimal 2-node network and a more biologically realistic 5-node network based on the *C. elegans* polarization system. They introduce a computational pipeline for systematically exploring reaction-diffusion network dynamics. Their study highlights the limitations of the widely used 2-node antagonistic network, demonstrating its susceptibility to simple modifications that disrupt polarization. However, they show that polarization stability can be restored by combining multiple regulatory mechanisms, and that spatially varying kinetic parameters can fine-tune the interface position. The authors further investigate the 5-node network of *C. elegans*, identifying key parameters that enhance its robustness against perturbations. Their findings provide novel insights into the mechanisms that ensure stable polarization in biological systems.

The major strengths of this work lie in its rigorous computational approach and the clarity of its findings. The authors demonstrate that the widely used 2-node antagonistic network is highly sensitive to parameter changes, requiring precise fine-tuning to maintain stable polarization. However, they show that stability can be restored through compensatory modifications, which expand the range of parameter sets supporting polarization. By further exploring spatial parameter variations, the authors reveal how compensatory adjustments can stabilize polarization patterns, offering insights into potential biological mechanisms regulating interface localization.

Extending their analysis to the *C. elegans* polarization network, the authors construct a 5-node model grounded in an extensive literature review. Their computational pipeline identifies key parameters that enhance robustness, and their model successfully replicates experimental observations, even in mutant conditions. Notably, among 34 possible network structures, only the naturally evolved 5-node network with mutual inhibition between specific components maintains stable polarization, highlighting its evolutionary optimization. This work significantly advances our understanding of polarization maintenance and provides a valuable framework for future in silico experiments.

Despite its strengths, the study has some limitations related to simplifying assumptions. The model neglects cortical flows and the role of actomyosin dynamics, which are known to be crucial during the establishment phase of polarization in the *C. elegans* zygote. While the authors focus on the maintenance phase, the absence of these biomechanical effects may limit the model's applicability to the full polarization process. Additionally, the assumption of infinitely fast cytoplasmic diffusion disregards potential effects of cytoplasmic flows on the stability of molecular distributions. Experimental measurements suggest that cytoplasmic diffusion coefficients are only an order of magnitude higher than membrane diffusion coefficients, meaning that finite diffusion combined with cytoplasmic flows could influence polarization stability. Although the authors acknowledge and discuss these limitations, incorporating these effects in future models could provide a more complete picture of the polarization dynamics in *C. elegans* embryos.

---

## [Author Response]

The following is the authors’ response to the original reviews

**eLife Assessment**
This manuscript makes valuable contributions to our understanding of cell polarisation dynamics and its underlying mechanisms. Through the development of a computational pipeline, the authors provide solid evidence that compensatory actions, whether regulatory or spatial, are essential for the robustness of the polarisation pattern. However, a more comprehensive validation against experimental data and a proper estimation of model parameters are required for further characterization and predictions in natural systems, such as the *C. elegans* embryo.

We sincerely thank the editor(s) for their pertinent assessment. We have carefully considered the constructive recommendations and made the necessary revisions in the manuscript, which are also detailed in this response letter. We have implemented most of the revisions requested by the reviewers. For the few requests we did not fully accept, we have provided justifications. The corresponding revisions in both the Manuscript and Supplementary Information are highlighted with a yellow background. To provide a more comprehensive validation against experimental data and model parameters used for characterizing and predicting natural systems, we reproduced the qualitative and semi-quantitative phenomenon in three more experimental groups previously published (Section 2.5; Fig. S8) [Gotta et al., Curr. Biol., 2001; Aceto et al., Dev. Biol., 2006]. Combined with the original experiments (Section 2.5; Fig. S7) [Hoege et al., Curr. Biol., 2010; Beatty et al., Development, 2010; Beatty et al., Development, 2013], now we have reproduced five experimental groups in total (two acting on LGL-1 and three on CDC-42), comprising eight perturbed conditions and using wild-type as the reference. These results effectively demonstrate how comprehensively the network structure and parameters capture the characteristics of the *C. elegans* embryo. We have also acknowledged the limitations of the current cell polarization model and provided, in 2. Results and 3. Discussion and conclusion, a detailed outline of potential model improvements.

**Joint Public Review:**
The polarisation phenomenon describes how proteins within a signalling network segregate into different spatial domains. This phenomenon holds fundamental importance in biology, contributing to various cellular processes such as cell migration, cell division, and symmetry breaking in embryonic morphogenesis. In this manuscript, the authors assess the robustness of stable asymmetric patterns using both a previously proposed minimal model of a 2-node network and a more realistic 5-node network based on the *C. elegans* cell polarisation network, which exhibits anterior-posterior asymmetry. They introduce a computational pipeline for numerically exploring the dynamics of a given reaction-diffusion network and evaluate the stability of a polarisation pattern. Typically, the establishment of polarisation requires the mutual inhibition of two groups of proteins, forming a 2-node antagonistic network. Through a reaction-diffusion formulation, the authors initially demonstrate that the widely-used 2-node antagonistic network for creating polarised patterns fails to maintain the polarised pattern in the face of simple modifications. However, the collapsed polarisation can be restored by combining two or more opposing regulations. The position of the interface can be adjusted with spatially varied kinetic parameters. Furthermore, the authors show that the 5-node network utilised by *C. elegans* is the most stable for maintaining polarisation against parameter changes, identifying key parameters that impact the position of the interface.

We sincerely thank the editor(s) for the pertinent summary!

While the results offer novel and insightful perspectives on the network's robustness for cell polarisation, the manuscript lacks comprehensive validation against experimental data, justified node-node network interactions, and proper estimation of model parameters (based on quantitative measurements or molecular intensity distributions). These limitations significantly restrict the utility of the model in making meaningful predictions or advancing our understanding of cell polarisation and pattern formation in natural systems, such as the *C. elegans* embryo.

We sincerely thank the editor(s) for the comment!

To provide a more comprehensive validation against experimental data and model parameters, we reproduced the qualitative and semi-quantitative phenomenon in three more experimental groups previously published (Section 2.5; Fig. S8) [Gotta et al., Curr. Biol., 2001; Aceto et al., Dev. Biol., 2006]. Combined with the original experiments (Section 2.5; Fig. S7) [Hoege et al., Curr. Biol., 2010; Beatty et al., Development, 2010; Beatty et al., Development, 2013], now we have reproduced five experimental groups in total (two acting on LGL-1 and three on CDC-42), comprising eight perturbed conditions and using wild-type as the reference. These meaningful predictions effectively demonstrate the utility of our model’s network structure and parameters in advancing our understanding of cell polarisation and pattern formation in natural systems, exemplified by the *C. elegans* embryo.

We have also acknowledged the limitations of the current cell polarization model and provided, in 2. Results and 3. Discussion and conclusion, a detailed outline of potential model improvements. The limitations include, but are not limited to, issues involving “node-node network interactions” and the “proper estimation of model parameters (based on quantitative measurements or molecular intensity distributions)”, both of which rely on experimental measurements of biological information. However, comprehensive experimental measurement data on every molecular species, their interactions, and each species’ intensity distribution in space and time were not fully available from prior research. Refinement is lacking for some of these interactions, potentially requiring years of additional experimentation. Moreover, for certain species at specific developmental stages, only relative (rather than absolute) intensity measurements are available. We agreed that such information is essential for establishing a more utilizable model and discussed it thoroughly in 3. Discussion and conclusion. From a theoretical perspective, we adopted assumptions from the previous literature and constructed a minimal model for a specific cell polarization phase to investigate the network's robustness, supported by five experimental groups and eight perturbed conditions in the *C. elegans* embryo.

The study extends its significance by examining how cells maintain pattern stability amid spatial parameter variations, which are common in natural systems due to extracellular and intracellular fluctuations. The authors found that in the 2-node network, varying individual parameters spatially disrupt the pattern, but stability is restored with compensatory variations. Additionally, the polarisation interface stabilises around the step transition between parameter values, making its localisation tunable. This suggests a potential biological mechanism where localisation might be regulated through signalling perception.

We sincerely thank the editor(s) for the pertinent review!

Focusing on the *C. elegans* cell polarisation network, the authors propose a 5-node network based on an exhaustive literature review, summarised in a supplementary table. Using their computational pipeline, they identify several parameter sets capable of achieving stable polarisation and claim that their model replicates experimental behaviour, even when simulating mutants. They also found that among 34 possible network structures, the wild-type network with mutual inhibition is the only one that proves viable in the computational pipeline. Compared with previous studies, which typically considered only 2- or 3-node networks, this analysis provides a more complete and realistic picture of the signalling network behind polarisation in the *C. elegans* embryo. In particular, the model for *C. elegans* cell polarisation paves the way for further in silico experiments to investigate the role of the network structure over the polarisation dynamics. The authors suggest that the natural 5-node network of *C. elegans* is optimised for maintaining cell polarisation, demonstrating the elegance of evolution in finding the optimal network structure to achieve certain functions.

We sincerely thank the editor(s) for the pertinent review!

Noteworthy limitations are also found in this work. To simplify the model for numerical exploration, the authors assume several reactions have equivalent dynamics, reducing the parameter space to three independent dimensions. While the authors briefly acknowledge this limitation in the "Discussion and Conclusion" section, further analysis might be required to understand the implications. For instance, it is not clear how the results depend on the particular choice of parameters. The authors showed that adding additional regulation might disrupt the polarised pattern, with the conclusion apparently depending on the strength of the regulation. Even for the 5-node wild-type network, which is the most robust, adding a strong enough self-activation of [A], as done in the 2-node network, will probably cause the polarised pattern to collapse as well.

We sincerely thank the editor(s) for the comment!

Now we have thoroughly expanded our acknowledgment of the model’s limitations in in 2. Results and 3. Discussion and conclusion. To rule out the equivalent dynamics assumption undermines our conclusions, we have added simulations showing that the cell polarization pattern stability does not depend on the exact strength of each regulation, provided the regulations on both sides are initially balanced as a whole (Fig. S5). Specifically, we used a Monte Carlo method to sample a wide range of various parameter values (*i.e.*, *γ*, *α*, *k*_1_, *k*_2_, *q*_1_, *q*_2_ and [X_c_]) for all nodes and regulations in simple 2-node network and *C. elegans* 5-node network, to achieve pattern stability. Under these conditions (*i.e.*, without any reduction in the parameter space), single-sided self-regulation, single-sided additional regulation, and unequal system parameters still cause the stable polarized pattern to collapse, consistent with our conclusions in the simplified conditions with the parameter space reduced to three independent dimensions.

Additionally, the authors utilise parameter values that are unrealistic, fail to provide units for some of them, and assume unknown parameter values without justification. The model appears to have non-dimensionalised length but not time, resulting in a mix of dimensional and non-dimensional variables that can be confusing. Furthermore, they assume equal values for Hill coefficients and many parameters associated with activation and inhibition pathways, while setting inhibition intensity parameters to 1. These arbitrary choices raise concerns about the fidelity of the proposed model in representing the real system, as their selected values could potentially differ by many orders of magnitude from the actual parameters.

We sincerely thank the editor(s) for the comment!

We apologize for the confusion. The non-dimensionalised parameter values are adopted from previous theoretical research [Seirin-Lee et al., Cells, 2020], which originates from the experimental measurement in [Goehring et al., J. Cell Biol., 2011; Goehring et al., Science, 2011]. With the *in silico* time set as 2 sec per step, now we have added the Supplemental Text justifying how the units are removed during non-dimensionalization. This demonstrates that the derived non-dimensionalized parameter in this paper achieves realistic values on the same order of magnitude as those observed in reality, confirming the fidelity of the proposed model in representing the real system.

The assumption of “equal values for Hill coefficients and many parameters associated with activation and inhibition pathways” is to reduce the parameter space for affordable computational cost. It is a widely-used strategy to fix Hill coefficients (Seirin-Lee et al., J. Theor. Biol., 2015; Seirin-Lee, Bull. Math. Biol., 2021) and unify parameter values for different pathways in network research about both cell polarization (Marée et al., Bull. Math. Biol., 2006; Goehring et al., Science, 2011; Trong et al., New J. Phys., 2014) and other biological topics (*e.g.*, plasmid transferring in the microbial community (Wang et al., Nat. Commun., 2020)), to control computational cost. Nevertheless, to rule out that the equivalent dynamics assumption undermines our conclusions, we have added simulations showing that the cell polarization pattern stability does not depend on the exact parameter values associated with activation and inhibition pathways, provided the regulations on both sides are initially balanced as a whole (Fig. S5). Specifically, we used a Monte Carlo method to sample a wide range of various parameter values (*i.e.*, *γ*, *α*, *k*_1_, *k*_2_, *q*_1_, *q*_2_ and [X_c_]) for all nodes and regulations in simple 2-node network and *C. elegans* 5-node network, to achieve pattern stability. Under these conditions (*i.e.*, without any reduction in the parameter space), single-sided self-regulation, single-sided additional regulation, and unequal system parameters still cause the stable polarized pattern to collapse, consistent with our conclusions in the simplified conditions with the parameter space reduced to three independent dimensions.

To confirm the fidelity of the proposed model in representing the real system, we reproduced the qualitative and semi-quantitative phenomenon in three more experimental groups previously published (Section 2.5; Fig. S8) [Gotta et al., *Curr. Biol.*, 2001; Aceto et al., *Dev. Biol.*, 2006]. Combined with the original experiments (Section 2.5; Fig. 5; Fig. S7) [Hoege et al., Curr. Biol., 2010; Beatty et al., Development, 2010; Beatty et al., Development, 2013], now we have reproduced five experimental groups in total (two acting on LGL-1 and three on CDC-42), comprising eight perturbed conditions and using wild-type as the reference. These results effectively demonstrate how comprehensively the network structure and parameters capture the characteristics of the *C. elegans* embryo. We have also acknowledged the limitations of the current cell polarization model and provided, in 2. Results and 3. Discussion and conclusion, a detailed outline of potential model improvements.

It is worth noting that, although a strict match between numerical and realistic parameter values with consistent units is always helpful, a lot of notable pure numerical studies successfully unveil principles that help interpret [Ma et al., Cell, 2009] and synthesize real biological systems [Chau et al., Cell, 2012]. These studies suggest that numerical analysis in biological systems remains powerful, even when comprehensive experimental data from prior research are not fully available.

The definition of stability and its evaluation in the proposed pipeline might also be too narrow. Throughout the paper, the authors discuss the stability of the polarised pattern, checked by an exhaustive search of the parameter space where the system reaches a steady state with a polarised pattern instead of a homogeneous pattern. It is not clear if the stability is related to the linear stability analysis of the reaction terms, as conducted in Goehring et al. (Science, 2011), which could indicate if a homogeneous state exists and whether it is stable or unstable. The stability test is performed through a pipeline procedure where they always start from a polarised pattern described by their model and observe how it evolves over time. It is unclear if the conclusions depend on the chosen initial conditions. Particularly, it is unclear what would happen if the initial distribution of posterior molecules is not exactly symmetric with respect to the anterior molecules, or if the initial polarisation is not strong.

We sincerely thank the editor(s) for the comment!

The definition of stability and its evaluation in the proposed pipeline consider two criteria: 1. The pattern is polarized; 2. The pattern is stable. Following simulations, figures, and videos (Fig. 1-6; Fig. S1-S5; Fig. S7-S9; Movie S1-S5) have sufficiently demonstrated that the parameters and networks set up capture the cell polarization dynamis regarding both the stable and unstable states very well.

Now we have added new simulation on alternative initial conditions. They demonstrating the necessity of a polarized initial pattern set up independently of the reaction-diffusion network during the establishment phase, probably through additional mechanisms such as the active actomyosin contractility and flow (Cuenca et al., Development, 2003; Gross et al., Nat. Phys., 2019). Our conclusions (*i.e.*, single-sided self-regulation, single-sided additional regulation, and unequal system parameters cause the stable polarized pattern to collapse) have little dependence on the chosen initial conditions as long as the unsymmetric initial patterns can set up a stable polarized pattern. A part of the simulations institutively show our conclusions still hold if the initial distribution of posterior molecules is not exactly symmetric with respect to the anterior molecules, or if the initial polarisation is not strong (Fig. S4 and Fig. S9).

Regarding the biological interpretation and relevance of the model, it overlooks some important aspects of the *C. elegans* polarisation system. The authors focus solely on a reaction-diffusion formulation to reproduce the polarisation pattern. However, the polarisation of the *C. elegans* zygote consists of two distinct phases: establishment and maintenance, with actomyosin dynamics playing a crucial role in both phases (see Munro et al., Dev Cell 2004; Shivas & Skop, MBoC 2012; Liu et al., Dev Biol 2010; Wang et al., Nat Cell Biol 2017). Both myosin and actin are crucial to maintaining the localisation of PAR proteins during cell polarisation, yet the authors neglect cortical flows during the establishment phase and any effects driven by myosin and actin in their model, failing to capture the system's complexity. How this affects the proposed model and conclusions about the establishment of the polarisation pattern needs careful discussion. Additionally, they assume that diffusion in the cytoplasm is infinitely fast and that cytoplasmic flows do not play any role in cell polarity. Finite cytoplasmic diffusion combined with cytoplasmic flows could compromise the stability of the anterior-posterior molecular distributions. The authors claim that cytoplasmic diffusion coefficients are two orders of magnitude higher than membrane diffusion coefficients, but they seem to differ by only one order of magnitude (Petrášek et al., Biophys. J. 2008). The strength of cytoplasmic flows has been quantified by a few studies, including Cheeks et al., and Curr Biol 2004.

We sincerely thank the editor(s) for the comment!

Indeed, previous research highlighted the importance of convective cortical flow in orchestrating the localisation of PAR proteins during the establishment phase of polarisation formation [Goehring et al., J. Cell Biol., 2011; Rose et al., WormBook, 2014; Beatty et al., Development, 2013]. However, during the maintenance phase, the non-muscle myosin II (NMY-2) is regulated downstream by the PAR protein network rather than serving as the primary upstream factor controlling PAR protein localization [Goehring et al., J. Cell Biol., 2011; Rose et al., WormBook, 2014; Beatty et al., Development, 2013]. While some theoretical studies integrated both reaction-diffusion dynamics and the effects of myosin and actin [Tostevin, 2008; Goehring, Science, 2011], others focused exclusively on reaction-diffusion dynamics [Dawes et al., Biophys. J., 2011; Seirin-Lee et al., Cells, 2020]. We have now clarified the distinction between the establishment and maintenance phases in 1. Introduction, emphasized our research focus on the reaction-diffusion dynamics during the maintenance phase in 2. Results, and provided a discussion of the omitted actomyosin dynamics to foster a more comprehensive understanding in the future in 3. Discussion and conclusion. The effect of the establishment phase is studied as the initial condition for the cell polarization simulation solely governed by reaction-diffusion dynamics, with new simulations demonstrating the necessity of a polarized initial pattern set up independently of the reaction-diffusion network during the establishment phase, probably through additional mechanisms such as the active actomyosin contractility and flow [Cuenca et al., Development, 2003; Gross et al., Nat. Phys., 2019].

Cytoplasmic and membrane diffusion coefficients differ by two orders of magnitude according to previous experimental measurements on PAR-2 and PAR-6 (Goehring et al., J. Cell Biol., 2011; Lim et al., Cell Rep., 2021). Many previous *C. elegans* cell polarization models have incorporated mass-conservation model combined with finite cytoplasmic diffusion, but this model description can lead to reverse spatial concentration distribution between the cell membrane and cytosol (Fig. 3 of Seirin-Lee et al., J. Theor. Biol., 2016; Fig. 2ab of Seirin-Lee et al., J. Math. Biol., 2020), disobeying experimental observation (Fig. 4A of Sailer et al., Dev. Cell, 2015; Fig. 1A of Lim et al., Cell Rep., 2021). This implies that the infinite cytoplasmic diffusion, without precise experiment-based parameter assignment or accounting for other hidden biological processes (*e.g.*, protein production and degradation), may be inappropriate in modeling the real spatial concentration distributions distinguished between the cell membrane and cytosol. To address this issue, some theoretical research incorporated protein production and degradation into their model, to acquire the consistent spatial concentration distribution between the cell membrane and cytosol (Tostevin et al., Biophys. J., 2008). More definitive experimental data on the spatiotemporal changes in protein diffusion, production, and degradation are essential for providing a more realistic representation of cellular dynamics and enhancing the model's predictive power.

Now we have acknowledged the possibly overlooked aspects of the *C. elegans* polarisation system in 3. Discussion and conclusion, a detailed outline of potential model improvements. Those aspects include, but are not limited to, issues involving “neglect cortical flows” and the “diffusion in the cytoplasm is infinitely fast”. From a theoretical perspective, we adopted assumptions from the previous literature and constructed a minimal model for a specific cell polarization phase to investigate the network's robustness. The meaningful predictions of five experimental groups and eight perturbed conditions in the *C. elegans* embryo faithfully supports the biological interpretation and relevance of the model.

Although the authors compare their model predictions to experimental observations, particularly in reproducing mutant behaviours, they do not explicitly show or discuss these comparisons in detail. Diffusion coefficients and off-rates for some PAR proteins have been measured (Goehring et al., JCB 2011), but the authors seem to use parameter values that differ by many orders of magnitude, perhaps due to applied scaling. To ensure meaningful predictions, whether their proposed model captures the extensive published data should be evaluated. Various cellular/genetic perturbations have been studied to understand their effects on anterior-posterior boundary positioning. Testing these perturbations' responses in the model would be important. For example, comparing the intensity distribution of PAR-6 and PAR-2 with measurements during the maintenance phase by Goehring et al., JCB 2011, or comparing the normalised intensity of PAR-3 and PKC-3 from the model with those measured by Wang et al., Nat Cell Biol 2017, during establishment and maintenance phases (in both wild-type and cdc-42 (RNAi) zygotes) could provide insightful validation. Additionally, in the presence of active CDC-42, it has been observed that PAR-6 extends further into the posterior side (Aceto et al., Dev Biol 2006). Conducting such validation tests is essential to convince readers that the model accurately represents the actual system and provides insights into pattern formation during cell polarisation.

We sincerely thank the editor(s) for the comment!

To provide more comprehensive validations and refinements to ensure the model accurately represents biological systems, we extensively reproduced the qualitative and semi-quantitative phenomenon in three more experimental groups previously published (Section 2.5; Fig. S8) [Gotta et al., Curr. Biol., 2001; Aceto et al., Dev. Biol., 2006]. Combined with the original experiments (Section 2.5; Fig. 5; Fig. S7) [Hoege et al., Curr. Biol., 2010; Beatty et al., Development, 2010; Beatty et al., Development, 2013], now we have reproduced five experimental groups in total from published data, comprising eight perturbed conditions and using wild-type as the reference. We have also explicitly show the comparison between model predictions and experimental observations (including the mutant behaviors reproduction as well) in detail, by describing how “cell polarization pattern characteristics in simulation” responds to various cellular/genetic perturbations (Section 2.5; Fig. 5; Fig. S7 and S8). The original and new validation tests conducted can convince readers that the model accurately represents the actual system and provides insights into pattern formation during cell polarisation.

The diffusion coefficients for anterior and posterior molecular species were assigned according to previous experimental and theoretical research (Goehring et al., J. Cell Biol., 2011; Goehring et al., Science, 2011; Seirin-Lee et al., Cells, 2020). The off-rates are assigned uniformly by searching viable parameter sets that can set up a network with cell polarization pattern stability. Now we have added simulations showing that the cell polarization pattern stability and response to network structure and parameter perturbation does not depend on the exact parameter values (*incl.*, diffusion coefficients and off-rates), provided the parameter values on both sides are initially balanced as a whole (Fig. S5). Specifically, we used a Monte Carlo method to sample a wide range of various parameter values (*i.e.*, *γ*, *α*, *k*_1_, *k*_2_, *q*_1_, *q*_2_ and [X_c_]) for all nodes and regulations in simple 2-node network and *C. elegans* 5-node network, to achieve pattern stability. Under these conditions (*i.e.*, without any reduction in the parameter space), single-sided self-regulation, single-sided additional regulation, and unequal system parameters still cause the stable polarized pattern to collapse, consistent with our conclusions in the simplified conditions with the parameter space reduced to three independent dimensions.

With the *in silico* time set as 2 sec per step, now we have added the Supplemental Text justifying how the units are removed during non-dimensionalization. This demonstrates that the derived non-dimensionalized parameter in this paper achieves realistic values on the same order of magnitude as those observed in reality, confirming the fidelity of the proposed model in representing the real system. We agreed that full experimental measurements of biological information are essential for establishing a more utilizable model and discussed it thoroughly in 3. Discussion and conclusion.

A clear justification, with references, for each network interaction between nodes in the five-node model is needed. Some of the activatory/inhibitory signals proposed by the authors have not been demonstrated (*e.g.*, CDC-42 directly inhibiting CHIN-1). Table S2 provided by the authors is insufficient to justify each node-node interaction, requiring additional explanations. (See the review by Gubieda et al., Phil. Trans. R. Soc. B 2020, for a similar node network that differs from the authors' model). Additionally, the intensity distributions of cortical PAR-3 and PKC-3 seem to vary significantly during both establishment and maintenance phases (Wang et al., Nat Cell Biol 2017), yet the authors consider the PAR-3/PAR-6/PKC-3 as a single complex. The choices in the model should be justified, as the presence or absence of clustering of these PAR proteins can be crucial during cell polarisation (Wang et al., Nat Cell Biol 2017; Dawes & Munro, Biophys J 2011).

We sincerely thank the editor(s) for the comment!

Now we have acknowledged the limitations of the current cell polarization model and provided, in 2. Results and 3. Discussion and conclusion, a detailed outline of potential model improvements. The limitations include, but are not limited to, issues involving “each network interaction between nodes” and the “consider the PAR-3/PAR-6/PKC-3 as a single complex”, in which the former one relies on experimental measurements of biological information. However, comprehensive experimental measurement data on every molecular species, their interactions, and each species’ intensity distribution in space and time were not fully available from prior research. Refinement is lacking for some of these interactions, potentially requiring years of additional experimentation. Moreover, for certain species at specific developmental stages, only relative (rather than absolute) intensity measurements are available. We agreed that such information is essential for establishing a more utilizable model and discussed it thoroughly in 3. Discussion and conclusion.

In consistent with previous modeling efforts [Goehring et al., Science, 2011; Gross et al., Nat. Phys., 2019; Lim et al., Cell Rep., 2021], our model treats the PAR-3/PAR-6/PKC-3 complex as a single entity for simplification, thus neglecting the potentially distinct spatial distributions of each single molecular species. We agree that a more comprehensive model, capable of resolving the individual localization patterns of these anterior PAR proteins, would be a valuable future direction. From a theoretical perspective, we adopted assumptions from the previous literature and constructed a minimal model for a specific cell polarization phase to investigate the network's robustness, supported by five experimental groups and eight perturbed conditions in the *C. elegans* embryo.

In summary, the authors successfully demonstrate the importance of compensatory actions in maintaining polarisation robustness. Their computational pipeline offers valuable insights into the dynamics of reaction-diffusion networks. However, the lack of detailed experimental validation and realistic parameter estimation limits the model's applicability to real biological systems. While the study provides a solid foundation, further work is needed to fully characterise and validate the model in natural contexts. This work has the potential to significantly impact the field by providing a new perspective on the robustness of cell polarisation networks.

We sincerely thank the editor(s) for the pertinent summary!

To provide a more comprehensive validation against experimental data and model parameters, three more groups of the qualitative and semi-quantitative phenomenon regarding CDC-42 are reproduced based on previously published experiments (Section 2.5; Fig. S8) [Gotta et al., Curr. Biol., 2001; Aceto et al., Dev. Biol., 2006]. Combined with the original experiments (Section 2.5; Fig. 5; Fig. S7) [Hoege et al., Curr. Biol., 2010; Beatty et al., Development, 2010; Beatty et al., Development, 2013], now we have reproduced five experimental groups in total, comprising eight perturbed conditions and using wild-type as the reference.

With the in silico time set as 2 sec per step, now we have added the Supplemental Text justifying how the units are removed during non-dimensionalization. This demonstrates that the derived non-dimensionalized parameter in this paper achieves realistic values on the same order of magnitude as those observed in reality, confirming the fidelity of the proposed model in representing the real system. Together with the reproduction of five experimental groups (eight perturbed conditions with wild-type as the reference), the model’s applicability to real biological systems in natural contexts are are fully characterised and validated.

The computational pipeline developed could be a valuable tool for further in silico experiments, allowing researchers to explore the dynamics of more complex networks. To maximise its utility, the model needs comprehensive validation and refinement to ensure it accurately represents biological systems. Addressing these limitations, particularly the need for more detailed experimental validation and realistic parameter choices, will enhance the model's predictive power and its applicability to understanding cell polarisation in natural systems.

We sincerely thank the editor(s) for the comment!

To provide more comprehensive validations and refinements to ensure the model accurately represents biological systems, we extensively reproduced the qualitative and semi-quantitative phenomenon in three more experimental groups previously published (Section 2.5; Fig. S8) [Gotta et al., Curr. Biol., 2001; Aceto et al., Dev. Biol., 2006]. Combined with the original experiments (Section 2.5; Fig. 5; Fig. S7) [Hoege et al., Curr. Biol., 2010; Beatty et al., Development, 2010; Beatty et al., Development, 2013], now we have reproduced five experimental groups in total from published data, comprising eight perturbed conditions and using wild-type as the reference. We have also explicitly show the comparison between model predictions and experimental observations (including the mutant behaviors reproduction as well) in detail, by describing how “cell polarization pattern characteristics in simulation” responds to various cellular/genetic perturbations (Section 2.5; Fig. 5; Fig. S7 and S8).

With the *in silico* time set as 2 sec per step, now we have added the Supplemental Text justifying how the units are removed during non-dimensionalization. This demonstrates that the derived non-dimensionalized parameter in this paper achieves realistic values on the same order of magnitude as those observed in reality, confirming the fidelity of the proposed model in representing the real system. Together with the reproduction of five experimental groups (eight perturbed conditions with wild-type as the reference), the model's predictive power and its applicability to understanding cell polarisation in natural systems are enhanced.

Now we have added simulations showing that the cell polarization pattern stability and response to network structure and parameter perturbation does not depend on the exact parameter values (*incl.*, diffusion coefficients, basal off-rates and inhibition intensity), provided the parameter values on both sides are initially balanced as a whole (Fig. S5). Specifically, we used a Monte Carlo method to sample a wide range of various parameter values (*i.e.*, *γ*, *α*, *k*_1_, *k*_2_, *q*_1_, *q*_2_ and [X_c_])for all nodes and regulations in simple 2-node network and *C. elegans* 5-node network, to achieve pattern stability. Under these conditions (*i.e.*, without any reduction in the parameter space), single-sided self-regulation, single-sided additional regulation, and unequal system parameters still cause the stable polarized pattern to collapse, consistent with our conclusions in the simplified conditions with the parameter space reduced to three independent dimensions.

**Recommendations for the Authors:**
(1) Parameterisation and Model Validation: The authors utilise parameter values that lack realism and fail to provide units for some of them, which can lead to confusion. For instance, the length of the cell is set to 0.5 without clear justification, raising questions about the scale used. Additionally, there's a mix of dimensional and non-dimensional variables, potentially complicating interpretation. Furthermore, arbitrary choices such as equal Hill coefficients and setting inhibition intensity parameters to 1 raise concerns about model fidelity. To ensure meaningful predictions, the authors should validate their model against extensive published data, including cellular/genetic perturbations. For example, comparing intensity distributions of PAR proteins measured during maintenance phases by Goehring et al., JCB 2011, and those obtained from the model could provide valuable validation. Similarly, comparisons with data from Wang et al., Nat Cell Biol 2017, on wild-type and cdc-42 (RNAi) zygotes, as well as observations from Aceto et al., Dev Biol 2006, on PAR-6 extension in the presence of active CDC-42, would strengthen the model's validity. Such validation tests are essential for convincing readers that the model accurately represents the actual system and can provide insights into pattern formation during cell polarisation.

We sincerely thank the editor(s) and referee(s) for the helpful suggestion!

Now we have added a new section, Parameter Nondimensionalization and Order of Magtitude Consistency, into Supplemental Text. In this section, we introduced how we adopted the parameter nondimensionalization and value assignments from previous works [Goehring et al., J. Cell Biol., 2011; Goehring et al., Science, 2011; Seirin-Lee et al., Cells, 2020]. We listed four examples (*i.e.*, evolution time, membrane diffusion coefficient, basal off-rate, and inhibition intensity) to show the consistency in order of magtitude between numerical and realistic values.

The assumption of “equal Hill coefficients” is to reduce the parameter space for an affordable computational cost. It is a widely-used strategy to fix Hill coefficients (Seirin-Lee et al., J. Theor. Biol., 2015; Seirin-Lee, Bull. Math. Biol., 2021) in network research, to control computational cost. Besides, setting inhibition intensity parameters to 1 is for determining a numerical scale. Now we have added simulations showing that the cell polarization pattern stability does not depend on the exact parameter values associated with activation and inhibition pathways, provided the regulations on both sides are initially balanced as a whole (Fig. S5). Specifically, we used a Monte Carlo method to sample a wide range of various parameter values (*i.e.*, *γ*, *α*, *k*_1_, *k*_2_, *q*_1_, *q*_2_ and [X_c_]) for all nodes and regulations in simple 2-node network and *C. elegans* 5-node network, to achieve pattern stability. Under these conditions (*i.e.*, without any reduction in the parameter space), single-sided self-regulation, single-sided additional regulation, and unequal system parameters still cause the stable polarized pattern to collapse, consistent with our conclusions in the simplified conditions with the parameter space reduced to three independent dimensions.

To confirm the fidelity of the proposed model in representing the real system, we reproduced the qualitative and semi-quantitative phenomenon in three more experimental groups previously published (Section 2.5; Fig. S8) [Gotta et al., Curr. Biol., 2001; Aceto et al., Dev. Biol., 2006]. Combined with the original experiments (Section 2.5; Fig. 5; Fig. S7) [Hoege et al., Curr. Biol., 2010; Beatty et al., Development, 2010; Beatty et al., Development, 2013], now we have reproduced five experimental groups in total (two acting on LGL-1 and three on CDC-42), comprising eight perturbed conditions and using wild-type as the reference. These results effectively demonstrate how comprehensively the network structure and parameters capture the characteristics of the *C. elegans* embryo. We have also acknowledged the limitations of the current cell polarization model and provided, in 2. Results and 3. Discussion and conclusion, a detailed outline of potential model improvements.

It is worth noting that, although a strict match between numerical and realistic parameter values with consistent units is always helpful, a lot of notable pure numerical studies successfully unveil principles that help interpret [Ma et al., Cell, 2009] and synthesize real biological systems [Chau et al., Cell, 2012]. These studies suggest that numerical analysis in biological systems remains powerful, even when comprehensive experimental data from prior research are not fully available.

(2) Parameter Changes: It is not clear how the parameters change as more complicated networks are explored, and how this affects the comparison between the simple and complete model. Clarification on this point would be beneficial.

We sincerely thank the editor(s) and referee(s) for the helpful suggestion!

The computational pipeline in Section 2.1 is generalized for all reaction-diffusion networks, including the simple and complete ones studied in this paper. The parameter changes included two parts: 1. The mutual activation in the anterior (none for the simple 2-node network and *q*<sub2 for the complete 5-node network); 2. The viable parameter sets (122 sets for the simple 2-node network and 602 sets for the complete 5-node network). Now we have explicitly clarified those differences:

Those differences don’t affect the comparison between the simple and complete models. Now we have added comprehensive comparisons between the simple and complete models about 1. How they respond to alternative initial conditions consistently (Fig. S2). 2. How they respond to alternative single modifications consistently (Fig. S4 and S9), even when the parameters (*i.e.*, *γ*, *α*, *k*_1_, *k*_2_, *q*_1_, *q*_2_ and [X_c_]) are assigned with various values concerning all nodes and regulations (Fig. S5).

(3) Exploration of Model Parameter Space: In the two-node dual antagonistic model, the authors observe that the cell polarisation pattern is unstable for different systems (Fig. 1). However, it remains uncertain whether this instability holds true for the entire model parameter space. Have the authors thoroughly screened the full model parameter space to support their statements? It would be beneficial for the authors to provide clarification on the extent of their exploration of the model parameter space to ensure the robustness of their conclusions.

We sincerely thank the editor(s) and referee(s) for the helpful suggestion!

The trade-off between considered parameter space and computational cost is a long-term challenge in network study as there are always numerous combinations of network nodes, edges, and parameters [Ma et al., Cell, 2009; Chau et al., Cell, 2012]. The computational pipeline in Section 2.1 generalized for all reaction-diffusion networks exerts two strategies to limit the computational cost and set up a basic network reference: 1. Dimension Reduction (Strategy 1) - Unifying the parameter values for different nodes and different edges within the same regulatory type to minimize the unidentical parameter numbers into 3; 2: Parameter Space Confinement (Strategy 2): Enumerating the dimensionless parameter set \begin{document}$\Omega\left(\gamma, k_{1}, q_{2}\right)$\end{document} on a three-dimensional (3D) grid confined by γ∈ [0,0.05] in steps ∆γ = 0.001, *k*_1_∈[0,5] in steps ∆*k*_1_ = 0.05, and in steps .

In the early stage of our project, we tried to explore “the entire model parameter space” as indicated by the reviewer. We first tried to use the Monte Carlo method to find parameter solutions in an open parameter space and with all parameter values allowed to be different. However, such a process is full of randomness and is computationally expensive (taking months to search viable parameter sets but still unable to profile the continuous viable parameter space; the probability of finding a viable parameter set is no higher than 0.02%, making it very hard to profile a continuous viable parameter space). Now we clearly can see the viable parameter space is a thin curved surface where all parameters have to satisfy a critical balance (Fig. 3a, b, Fig. 5e, f). This is why we exert a typical strategy for dimension reduction in network research in both cell polarization [Marée et al., Bull. Math. Biol., 2006; Goehring et al., Science, 2011; Trong et al., New J. Phys., 2014] and other biological topics (*e.g.*, plasmid transferring in the microbial community [Wang et al., *Nat. Commun.*, 2020]), *i.e.*, unifying the parameter values for different nodes and different edges within the same regulatory type.

Additionally, the curved surface for viable parameter space can be extended to infinite as long as the parameter balance is achieved (Fig. 3a, b, Fig. 5e, f), it is impossible or unnecessary to explore “the entire model parameter space”. Setting up a confined parameter region near the original point for parameter enumeration can help profile the continuous viable parameter space, which is sufficient for presenting the central conclusion of this paper – that is - the network structure and parameter need to satisfy a balance for stable cell polarization.

To support a comprehensive study considering all kinds of reference and perturbed networks, we have maximized the parameter domain size by exhausting all the computational research we can access, including 400-500 Intel(R) Core(TM) E5-2670v2 and Gold 6132 CPU on the server (High-Performance Computing Platform at Peking University) and 5 Intel(R) Core(TM) i9-14900HX CPU on personal computers.

To make it certain that instability holds true when the model parameter space is extended, we add a comprehensive comparison between the simple and complete models about how their instability occurs consistently even when the parameters (*i.e.*, *γ*, *α*, *k*_1_, *k*_2_, *q*_1_, *q*_2_ and [X_c_])are assigned with various values concerning all nodes and regulations, searched by the Monte Carlo method (Fig. S5).

(4) Sensitivity of Numerical Solutions to Initial Conditions: Are the numerical solutions in both models sensitive to the chosen initial condition? What results do the models provide if uniform initial distributions were utilised instead?

We sincerely thank the editor(s) and referee(s) for the comments!

To investigate both the simple network and the realistic network consisting of various node numbers and regulatory pathways [Goehring et al., Science, 2011; Lang et al., Development, 2017], we propose a computational pipeline for numerical exploration of the dynamics of a given reaction-diffusion network's dynamics, specifically targeting the maintenance phase of stable cell polarization after its initial establishment [Motegi et al., Nat. Cell Biol., 2011; Goehring et al., Science, 2011; Seirin-Lee et al., Cells, 2020].

Now we have added new simulations and explanations for the sensitivity of numerical solutions to initial conditions. For both models, a uniform initial distribution leads to a homogeneous pattern while a Gaussian noise distribution leads to a multipolar pattern. In contrast, an initial polarized distribution (even with shifts in transition planes, weak polarization, or asymmetric curve shapes between the two molecular species) can maintain cell polarization reliably.

(5) Initial Conditions and Stability Tests: In Figure 1, the authors discuss the stability of the basic two-node network (a) upon modifications in (b-d). The stability test is performed through a pipeline procedure in which they always start from a polarised pattern described by Equation (4) and observe how the pattern evolves over time. It would be beneficial to explore whether the stability test depends on this specific initial condition. For instance, what would happen if the posterior molecules have an initial distribution of 1/(1+e^(-10x)), which is not exactly symmetric with respect to the anterior molecules' distribution of 1-1/(1+e^(-20x))? Additionally, if the initial polarisation is not as strong, for example, with the anterior molecules having a distribution of 10-1/(1+e^(-20x)) and the posterior molecules having a distribution of 9+1/(1+e^(-20x)), how would this affect the results?

We sincerely thank the editor(s) and referee(s) for the constructive advice!

Now we have added comprehensive comparisons between the simple and complete models about how they respond to alternative initial conditions consistently (Fig. S4, Fig. S9). The successful cell polarization pattern requests an initial polarized pattern, but its following stability and response to perturbation depend very little on the specific form of the initial polarized pattern. All the conditions mentioned by the reviewer have been included.

(6) Stability Analysis: Throughout the paper, the authors discuss the stability of the polarised pattern. The stability is checked by an exhaustive search of the parameter space, ensuring the system reaches a steady state with a polarised pattern instead of a homogeneous pattern. It would be beneficial to explore if this stability is related to a linear stability analysis of the model parameters, similar to what was conducted in Reference [18], which can determine if a homogeneous state exists and whether it is stable or unstable. Including such an analysis could provide deeper insights into the system's stability and validate its robustness.

We sincerely thank the editor(s) and referee(s) for the comments!

We agree that the linear stability analysis can potentially offer additional insights into polarized pattern behavior. However, this approach often requests the aid of numerical solutions and is therefore not entirely independent [Goehring et al., Science, 2011]. Over the past decade, numerical simulations have consistently proven to be a reliable and sufficient approach for studying network dynamics, spanning from *C. elegans* cell polarization [Tostevin et al., *Biophys. J*, 2008; Blanchoud et al., Biophys. J, 2015; Seirin-Lee, Dev. Growth Differ., 2020] to topics in metazoon [Chau et al., Cell, 2012; Qiao et al., eLife, 2022; Sokolowski et al., arXiv, 2023]. Numerous purely numerical studies have successfully unveiled principles that help interpret [Ma et al., Cell, 2009] and synthesized real biological systems [Chau et al., Cell, 2012], independent of additional mathematical analysis. Thus, we leverage our numerical framework to address the cell polarization problems cell polarization problems in this paper.

To confirm the reliability of stability checked by an exhaustive search of the parameter space, now we reproduce the qualitative and semi-quantitative phenomenon in three more experimental groups previously published (Section 2.5; Fig. S8) [Gotta et al., Curr. Biol., 2001; Aceto et al., Dev. Biol., 2006]. Combined with the original experiments (Section 2.5; Fig. 5; Fig. S7) [Hoege et al., Curr. Biol., 2010; Beatty et al., Development, 2010; Beatty et al., Development, 2013], we reproduce five experimental groups in total (two acting on LGL-1 and three acting on CDC-42), comprising eight perturbed conditions and using wild-type as the reference.

To confirm the robustness of our conclusions regarding the system's stability, now we add comprehensive comparisons between the simple and complete models about 1. How they respond to alternative initial conditions consistently (Fig. S4; Fig. S9). 2. How they respond to alternative single modifications consistently, even when the parameters (*i.e.*, *γ*, *α*, *k*_1_, *k*_2_, *q*_1_, *q*_2_ and [X_c_]) are assigned with various values concerning all nodes and regulations (Fig. S5).

(7) Interface Position Determination: In Figure 4, the authors demonstrate that by using a spatially varied parameter, the position of the interface can be tuned. Particularly, the interface is almost located at the step where the parameter has a sharp jump. However, in the case of a homogeneous parameter (*e.g.*, Figure 4(a)), the system also reaches a stable polarised pattern with the interface located in the middle (x = 0), similar to Figure 4(b), even though the homogeneous parameter does not contain any positional information of the interface. It would be helpful to clarify the difference between Figure 4(a) and Figure 4(b) in terms of the interface position determination.

We sincerely thank the editor(s) and referee(s) for the comments!

The case of a homogeneous parameter (*e.g.*, Fig. 4a), in which the system also reaches a stable polarised pattern with the interface located in the middle (x = 0), is just a reference adopted from Fig. 1a to show that the inhomogeneous positional information in Fig. 4b can achieve a similar stable polarised pattern.

Now we clarify the interface position determination to Section 2.4 to improve readability. Moreover, it is marked with grey dashed line in all the patterns in Fig. 4 and Fig. 6 to highlight the importance of inhomogeneous parameters on interface localization.

(8) Presented Comparison with Experimental Observations: The comparison with experimental observations lacks clarity. It isn't clear that the model "faithfully recapitulates" the experimental observations (lines 369-370). We recommend discussing and showing these comparisons more carefully, highlighting the expectations and similarities.

We sincerely thank the editor(s) and referee(s) for the constructive suggestion!

Now we remove the word “faithfully” and highlight the expectations and similarities of each experimental group by describing “cell polarization pattern characteristics in simulation: …”.

(9) Validation of Model with Experimental Data: Given the extensive number of model parameters and the uncertainty of their values, it is essential for the authors to validate their model by comparing their results with experimental data. While *C. elegans* polarisation has been extensively studied, the authors have yet to utilise existing data for parameter estimation and model validation. Doing so would considerably strengthen their study.

We sincerely thank the editor(s) and referee(s) for the constructive suggestion!

To utilise existing data for parameter estimation, now we add a new section, Parameter Nondimensionalization and Order of Magtitude Consistency, into Supplemental Text. In this section, we introduced how we adopted the parameter nondimensionalization and value assignments from previous works [Goehring et al., J. Cell Biol., 2011; Goehring et al., Science, 2011; Seirin-Lee et al., Cells, 2020]. We listed four examples (*i.e.*, evolution time, membrane diffusion coefficient, basal off-rate, and inhibition intensity) to show the consistency in order of magtitude between numerical and realistic values.

To utilise existing data for model validation, now we reproduce the qualitative and semi-quantitative phenomenon in three more experimental groups previously published (Section 2.5; Fig. S8) [Gotta et al., Curr. Biol., 2001; Aceto et al., Dev. Biol., 2006]. Combined with the original experiments (Section 2.5; Fig. 5; Fig. S7) [Hoege et al., Curr. Biol., 2010; Beatty et al., Development, 2010; Beatty et al., Development, 2013], we reproduce five experimental groups in total (two acting on LGL-1 and three acting on CDC-42), comprising eight perturbed conditions and using wild-type as the reference.

Also, we acknowledge the limitations of the current cell polarization model and provided, in 3. Discussion and conclusion, a detailed outline of potential model improvements. The limitations include, but are not limited to, issues involving “extensive number of model parameters” and “uncertainty of their values”, both of which rely on experimental measurements of biological information. However, comprehensive experimental measurement data on every molecular species, their interactions, and each species’ intensity distribution in space and time were not fully available from prior research. Refinement is lacking for some of these interactions, potentially requiring years of additional experimentation. Moreover, for certain species at specific developmental stages, only relative (rather than absolute) intensity measurements are available. We agreed that such information is essential for establishing a more utilizable model and discussed it thoroughly in 3. Discussion and conclusion. From a theoretical perspective, we adopted assumptions from the previous literature and constructed a minimal model for a specific cell polarization phase to investigate the network's robustness, supported by five experimental groups and eight perturbed conditions with wild-type as a reference in the *C. elegans* embryo.

(10) Enhancing Model Accuracy by Considering Cortical Flows: The authors are encouraged to include cortical flows in their cell polarisation model, as these flows are known to be pivotal in the process. Although the current model successfully predicts cell polarisation without accounting for cortical flows, research has demonstrated their significant role in polarisation formation. By incorporating cortical flows, the model would provide a more thorough and precise representation of the biological process. Furthermore, previous studies, such as those by Goehring et al. (References 17 and 18), highlight the importance of convective actin flow in initiating polarisation. It would be valuable for the authors to address the contribution of convection with actin flow to the establishment of the polarisation pattern. The polarisation of the *C. elegans* zygote progresses through two distinct phases: establishment and maintenance, both heavily influenced by actomyosin dynamics. Works by Munro et al. (Dev Cell 2004), Shivas & Skop (MBoC 2012), Liu et al. (Dev. Biol. 2010), and Wang et al. (Nat Cell Biol 2017) underscore the critical roles of myosin and actin in orchestrating the localisation of PAR proteins during cell polarisation. To enhance the fidelity of their model, we recommend that the authors either integrate cortical flows and consider the effects driven by myosin and actin, or provide a discussion on the repercussions of omitting these dynamics.

We sincerely thank the editor(s) and referee(s) for the comment!

Indeed, previous research highlighted the importance of convective cortical flow in orchestrating the localisation of PAR proteins during the establishment phase of polarisation formation [Goehring et al., J. Cell Biol., 2011; Rose et al., WormBook, 2014; Beatty et al., Development, 2013]. However, during the maintenance phase, the non-muscle myosin II (NMY-2) is regulated downstream by the PAR protein network rather than serving as the primary upstream factor controlling PAR protein localization. While some theoretical studies integrated both reaction-diffusion dynamics and the effects of myosin and actin [Tostevin et al., Biophys J, 2008; Goehring et al, Science, 2011], others focused exclusively on reaction-diffusion dynamics [Dawes et al., Biophys. J., 2011; Seirin-Lee et al., Cells, 2020]. Now we clarify the distinction between the establishment and maintenance phases, emphasize our research focus on the reaction-diffusion dynamics during the maintenance phase, and provide a discussion of these omitted dynamics to foster a more comprehensive understanding in the future, as suggested.

(11) Further Justification of Network Interactions: The authors should provide additional explanations, supported by empirical evidence, for the network interactions assumed in their model. This includes both node-node interactions and the rationale behind protein complex formations. Some of the proposed interactions lack empirical validation, as noted in studies such as Gubieda et al., Phil. Trans. R. Soc. B 2020. Additionally, discrepancies in protein intensity distributions, as observed in Wang et al., Nat Cell Biol 2017, should be addressed, particularly concerning the consideration of the PAR-3/PAR-6/PKC-3 complex as a single entity. Justifying these choices is crucial for ensuring the model's credibility and alignment with experimental findings.

We sincerely thank the editor(s) and referee(s) for the helpful advice!

In consistency with previous modeling efforts [Goehring et al., Science, 2011; Gross et al., Nat. Phys., 2019; Lim et al., Cell Rep., 2021], our model treats the PAR-3/PAR-6/PKC-3 complex as a single entity for simplification, thus neglecting the potentially distinct spatial distributions of each single molecular species.

Now we acknowledge the limitations of the current cell polarization model and provided, in 3. Discussion and conclusion, a detailed outline of potential model improvements. The limitations include, but are not limited to, issues involving “node-node interactions” and “discrepancies in protein intensity distributions”, both of which rely on experimental measurements of biological information. However, comprehensive experimental measurement data on every molecular species, their interactions, and each species’ intensity distribution in space and time were not fully available from prior research. Refinement is lacking for some of these interactions, potentially requiring years of additional experimentation. Moreover, for certain species at specific developmental stages, only relative (rather than absolute) intensity measurements are available. We agreed that such information is essential for establishing a more utilizable model and discussed it thoroughly in 3. Discussion and conclusion.

To ensure the model's credibility and alignment with experimental findings, now we reproduce the qualitative and semi-quantitative phenomenon in three more experimental groups previously published (Section 2.5; Fig. S8) [Gotta et al., Curr. Biol., 2001; Aceto et al., Dev. Biol., 2006]. Combined with the original experiments (Section 2.5; Fig. 5; Fig. S7) [Hoege et al., Curr. Biol., 2010; Beatty et al., Development, 2010; Beatty et al., Development, 2013], now we have reproduced five experimental groups in total (two acting on LGL-1 and three on CDC-42), comprising eight perturbed conditions and using wild-type as the reference.

(12) Further Justification of Node-Node Network Interactions: The authors should provide further justification for the node-node network interactions assumed in their study. To the best of our knowledge, some of the node-node interactions proposed have not yet been empirically demonstrated. Providing additional explanations for these interactions would enhance the credibility of the model and ensure its alignment with empirical evidence.

We sincerely thank the editor(s) and referee(s) for the helpful advice!

Now we acknowledge the limitations of the current cell polarization model and provided, in 3. Discussion and conclusion, a detailed outline of potential model improvements. The limitations include, but are not limited to, issues involving “node-node network interactions”, which rely on experimental measurements of biological information. However, comprehensive experimental measurement data on every molecular species, their interactions, and each species’ intensity distribution in space and time were not fully available from prior research. Refinement is lacking for some of these interactions, potentially requiring years of additional experimentation. Moreover, for certain species at specific developmental stages, only relative (rather than absolute) intensity measurements are available. We agreed that such information is essential for establishing a more utilizable model and discussed it thoroughly in 3. Discussion and conclusion.

To enhance the credibility of the model and ensure its alignment with empirical evidence, we reproduced the qualitative and semi-quantitative phenomenon in three more experimental groups previously published (Section 2.5; Fig. S8) [Gotta et al., Curr. Biol., 2001; Aceto et al., Dev. Biol., 2006]. Combined with the original experiments (Section 2.5; Fig. 5; Fig. S7) [Hoege et al., Curr. Biol., 2010; Beatty et al., Development, 2010; Beatty et al., Development, 2013], now we have reproduced five experimental groups in total (two acting on LGL-1 and three on CDC-42), comprising eight perturbed conditions and using wild-type as the reference.

(13) Justification for Network Interactions and Protein Complexes: The authors must provide clear justifications, supported by references, for each network interaction between nodes in the five-node model. Some of the activatory/inhibitory signals proposed lack empirical validation, such as CDC-42 directly inhibiting CHIN-1. The provided Table S2 is insufficient to justify these interactions, necessitating additional explanations. Reviewing relevant literature, such as the work by Gubieda et al., Phil. Trans. R. Soc. B 2020, may offer insights into similar node networks. Furthermore, the authors should address discrepancies in protein intensity distributions, as observed in studies like Wang et al., Nat Cell Biol 2017. Specifically, the authors consider the PAR-3/PAR-6/PKC-3 complex as a single entity despite potential differences in their distributions. Justification for this choice is essential, particularly considering the importance of clustering dynamics during cell polarisation, as demonstrated by Wang et al., Nat Cell Biol 2017, and Dawes & Munro, Biophys J 2011.

We sincerely thank the editor(s) and referee(s) for the helpful advice!

In consistent with previous modeling efforts [Goehring et al., Science, 2011; Gross et al., Nat. Phys., 2019; Lim et al., Cell Rep., 2021], our model treats the PAR-3/PAR-6/PKC-3 complex as a single entity for simplification, thus neglecting the potentially distinct spatial distributions of each single molecular species. Besides, the inhibition of CHIN-1 from CDC-42, which recruits cytoplasmic PAR-6/PKC-3 to form a complex, may act indirectly to restrict CHIN-1 localization through phosphorylation [Sailer et al., Dev. Cell, 2015; Lang et al., Development, 2017].

Now we acknowledge the limitations of the current cell polarization model and provided, in 3. Discussion and conclusion, a detailed outline of potential model improvements. The limitations include, but are not limited to, issues involving “each network interaction between nodes in the five-node model” and “discrepancies in protein intensity distributions”, both of which rely on experimental measurements of biological information. However, comprehensive experimental measurement data on every molecular species, their interactions, and each species’ intensity distribution in space and time were not fully available from prior research. Refinement is lacking for some of these interactions, potentially requiring years of additional experimentation. Moreover, for certain species at specific developmental stages, only relative (rather than absolute) intensity measurements are available. We agreed that such information is essential for establishing a more utilizable model and discussed it thoroughly in 3. Discussion and conclusion. From a theoretical perspective, we adopted assumptions from the previous literature and constructed a minimal model for a specific cell polarization phase to investigate the network's robustness, supported by five experimental groups and eight perturbed conditions with wild-type as a reference in the *C. elegans* embryo.

(14) Incorporating Cytoplasmic Dynamics into the Model: The authors assume infinite cytoplasmic diffusion and neglect the role of cytoplasmic flows in cell polarity, which may oversimplify the model. Finite cytoplasmic diffusion combined with flows could potentially compromise the stability of anterior-posterior molecular distributions, affecting the accuracy of the model's predictions. The authors claim a significant difference between cytoplasmic and membrane diffusion coefficients, but the actual disparity seems smaller based on data from Petrášek et al., Biophys. J. 2008. For example, cytosolic diffusion coefficients for NMY-2 and PAR-2 differ by less than one order of magnitude. Additionally, the strength of cytoplasmic flows, as quantified by studies such as Cheeks et al., and Curr Biol 2004, should be considered when assessing the impact of cytoplasmic dynamics on polarity stability. Incorporating finite cytoplasmic diffusion and cytoplasmic flows into the model could provide a more realistic representation of cellular dynamics and enhance the model's predictive power.

We sincerely thank the editor(s) and referee(s) for the comment!

Cytoplasmic and membrane diffusion coefficients differ by two orders of magnitude according to previous experimental measurements on PAR-2 and PAR-6 (Goehring et al., J. Cell Biol., 2011; Lim et al., Cell Rep., 2021). Many previous *C. elegans* cell polarization models have incorporated mass-conservation model combined with finite cytoplasmic diffusion, but this model description can lead to reverse spatial concentration distribution between the cell membrane and cytosol (Fig. 3 of Seirin-Lee et al., J. Theor. Biol., 2016; Fig. 2ab of Seirin-Lee et al., J. Math. Biol., 2020), disobeying experimental observation (Fig. 4A of Sailer et al., Dev. Cell, 2015; Fig. 1A of Lim et al., Cell Rep., 2021). This implies that the infinite cytoplasmic diffusion, without precise experiment-based parameter assignment or accounting for other hidden biological processes (*e.g.*, protein production and degradation), may be inappropriate in modeling the real spatial concentration distributions distinguished between the cell membrane and cytosol. To address this issue, some theoretical research incorporated protein production and degradation into their model, to acquire the consistent spatial concentration distribution between the cell membrane and cytosol (Tostevin et al., Biophys. J., 2008). More definitive experimental data on the spatiotemporal changes in protein diffusion, production, and degradation are essential for providing a more realistic representation of cellular dynamics and enhancing the model's predictive power.

Cytoplasmic flows indeed play an unneglectable role in cell polarity during the establishment phase [Kravtsova et al., Bull. Math. Biol., 2014], which creates a spatial gradient of actomyosin contractility and directs PAR-3/PKC-3/PAR-6 to the anterior membrane by cortical flow [Rose et al., WormBook, 2014; Lang et al., Development, 2017]. However, during the maintenance phase, the non-muscle myosin II (NMY-2) is regulated downstream by the PAR protein network rather than serving as the primary upstream factor controlling PAR protein localization [Goehring et al., J. Cell Biol., 2011; Rose et al., WormBook, 2014; Geβele et al., Nat. Commun., 2020]. While some theoretical studies integrated both reaction-diffusion dynamics and the effects of myosin and actin [Tostevin, 2008; Goehring, Science, 2011], others focused exclusively on reaction-diffusion dynamics [Dawes et al., Biophys. J., 2011; Seirin-Lee et al., Cells, 2020]. We now emphasize our research focus on the reaction-diffusion dynamics during the maintenance phase, so the dynamics between NMY-2 and PAR-2 are not included. We have also provided a discussion of the simplified cytoplasmic diffusion and omitted cytoplasmic flows to foster a more comprehensive understanding in the future.

(15) Explanation of Lethality References: On page 13, the authors mention lethality without adequately explaining why they are drawing connections with lethality experimental data.

We sincerely thank the editor(s) and referee(s) for the comment!

It is well-known that cell polarity loss in *C. elegans* zygote will lead to symmetric cell division, which brings out the more symmetric allocation of molecular-to-cellular contents in daughter cells; this will result in abnormal cell size, cell cycle length, and cell fate in daughter cells, followed by embryo lethality [Beatty et al., Development, 2010; Beatty et al., Development, 2013; Rodriguez et al., Dev. Cell, 2017; Jankele et al., eLife, 2021]. Now we explain why we are drawing connections with lethality experimental data in Section 2.5.

(16) Improved Abstract: "...However, polarity can be restored through a combination of two modifications that have opposing effects..." This sentence could be revised for better clarity. For example, the authors could consider rephrasing it as follows: "...However, polarity restoration can be achieved by combining two modifications with opposing effects...".

We sincerely thank the editor(s) and referee(s) for helpful advice!

Now we revise the abstract as follows:

“Abstract – However, polarity restoration can be achieved by combining two modifications with opposing effects.”

(17) Conservation of Mass in Network Models: Is conservation of mass satisfied in their network models?

We sincerely thank the editor (s) and referee(s) for the comment!

While previous experiments provide evidence for near-constant protein mass during the establishment phase [Goehring et al., Science, 2011], whether this is consistent until the end of maintenance is unclear.

Many previous *C. elegans* cell polarization models have assumed mass conservation on the cell membrane and in the cell cytosol, this model description can lead to reverse spatial concentration distribution between the cell membrane and cytosol [Fig. 3 of Seirin-Lee et al., J. Theor. Biol., 2016; Fig. 2ab of Seirin-Lee et al., J. Math. Biol., 2020], disobeying experimental observation [Fig. 4A of Sailer et al., Dev. Cell, 2015; Fig. 1A of Lim et al., Cell Rep., 2021]. This implies that mass conservation may be inappropriate in modeling the real spatial concentration distributions distinguished between the cell membrane and cytosol. To address this issue, some theoretical research incorporated protein production and degradation into their model, instead of assuming mass conservation [Tostevin et al., Biophys. J., 2008]. More definitive experimental data on the spatiotemporal changes in protein mass are essential for constructing a more accurate model.

Given the absence of a universally accepted model in agreement with experimental observation, we adopted the assumption that the concentration of molecules in the cytosol (not the total mass on the cell membrane and in the cell cytosol) is spatially inhomogeneous and temporally constant, which was also used before [Kravtsova et al., Bull. Math. Biol., 2014]. In the context of this well-mixed constant cytoplasmic concentration, our model successfully reproduced the cell polarization phenotype in wild-type and eight perturbed conditions (Section 2.5; Fig. S7; Fig. S8), supporting the validity of this simplified, yet effective, model. Now we have provided a discussion of protein mass assumption to foster a more comprehensive understanding in the future.

(18) Comparison of Network Structures: In Figure 1c, the authors demonstrate that the symmetric two-node network is susceptible to single-sided additional regulation. They considered four subtypes of modifications, depending on whether [L] is in the anterior or posterior and The difference between the structures where whether [A] and [L] are mutually activating or inhibiting. What is the difference between the structure where [L] is in the anterior and in the posterior? Upon comparing the time evolution of the left panel ([L] is sided with) and the right panel ([L] is sided with [A]), the difference is so tiny that they are almost indistinguishable. It might be beneficial for the authors to provide a clearer explanation of the differences between these network structures to aid in understanding their implications.

We sincerely thank the editor(s) and referee(s) for the constructive suggestion!

The difference between the structures where [L] is in the anterior and posterior is the initial spatial concentration distribution of [L], which is polarized to have a higher concentration in the anterior and posterior respectively. The time evolution of the left panel ([L] is sided with [P]) and the right panel [L] is sided with [P] is almost indistinguishable because the perturbation from [L] is slight (less than over one order of magnitude) compared to the predominant [A]~[P] interaction \begin{document}$k_{\mathrm{A} 2}^{\mathrm{P}}=k_{\mathrm{P} 2}^{\mathrm{A}}=1$\end{document} for [A]~[P] mutual inhibition while for [A]~[L] mutual inhibition and *t*=0,100, 200, 300, 400 and 500 in Fig. S3, where the difference between the [L] ones in the left and right panels are distinguishably shown \begin{document}$q_{\mathrm{L} 2}^{\mathrm{A}} \& q_{\mathrm{A} 2}^{\mathrm{L}}=0.012$\end{document} for [A]~[L] mutual activation, highlighting the response of cell polarization pattern. To aid the readers in understanding their implications, we have added the [L] and plotted the spatial concentration distribution of all three molecular species at

(19) Figure Reference: In line 308, Fig. 4a is referenced when explaining the loss of pattern stability by modifying an individual parameter, but this is not shown in that panel. Please update the panel or adjust the reference in the main text.

We sincerely thank the editor(s) and referee(s) for pointing out this problem!

Fig. 4 focuses on the regulatable shift of the zero-velocity interface by modifying a pair of individual parameters, not on the loss (or recovery) of pattern stability, which has been analyzed as a focus in Fig. 1, Fig. 2, and Fig. 3. Fig. 4a is actually from the same simulation as the one in Fig. 1a, which has spatially uniform parameters used as a reference in Fig. 4. The individual parameter modification in other subfigures of Fig. 4 shows how the zero-velocity interface is shifted in a regulatable manner always in the context of pattern stability. Now we update the panel, adjust the reference, add one more paragraph, and improve the wording to clarify how the analyses in Fig. 4 are carried out on top of the pattern stability already studied.

(20) Viable Parameter Sets: In line 355, the number of viable parameter sets (602) is not very informative by itself. We suggest reporting the fraction or percentage of sets tested that resulted in viable results instead. This applies similarly to lines 411 and 468.

We sincerely thank the editor(s) and referee(s) for the constructive comment!

Now the fraction/percentage of parameter sets tested that resulted in viable results are added everywhere the number appears.

(21) Perturbation Experiments: In lines 358-359, "the perturbation experiments" implies that those considered are the only possible ones. Please rephrase to clarify.

We sincerely thank the editor(s) and referee(s) for the helpful advice!

Now we rephrase three paragraphs to clarify why the perturbation experiments involved with [L] and [C] are considered instead of other possible ones.

(22) Figure 2S: This figure is unclear. The caption states that panel (a) shows the "final concentration distribution," but only a line is shown. If "distribution" refers to spatial distribution, please clarify which parameters are shown.

We sincerely thank the editor(s) and referee(s) for pointing out this problem!

Now we clarify the “spatial concentration distribution” and which parameters are shown in the figure caption.

(23) Figure 5 and 6 Captions: The captions for Figures 5 and 6 could benefit from clarification for better understanding.

We sincerely thank the editor(s) and referee(s) for the constructive suggestion!

Now we clarify the details in the captions of Fig. 5 and Fig. 6 for better understanding.

(24) Figure 5 Legend: The legend on the bottom right corner of Figure 5 is unclear. Please specify to which panel it refers.

We sincerely thank the editor(s) and referee(s) for the constructive suggestion!

Now we clarify to which the legend on the bottom right corner of Fig. 5 refers.

(25) L and A~C Interactions: In paragraphs 405-418, please explain why the L and A~C interactions are removed for the comparison instead of others.

We sincerely thank the editor(s) and referee(s) for the constructive suggestion!

Now we add a separate paragraph and a supplemental figure to explain why the L and A~C interactions are removed for the comparison instead of others.

(26) Network Structures in Figure S3: From the "34 possible network structures" considered in Figure S3 (lines 440-441), why are the "null cases" (L disconnected from the network) relevant? Shouldn't only 32 networks be considered?

We sincerely thank the editor(s) and referee(s) for pointing out this problem!

Now the two “null cases” are removed:

(27) Figure S3 Caption: The caption must state that the position of the nodes (left or right) implies the polarisation pattern. Additionally, with the current size of the figure, the dashed lines are extremely hard to differentiate from the continuous lines.

We sincerely thank the editor(s) and referee(s) for the constructive suggestion!

Now we state that the position of the nodes (left or right) implies the polarization pattern. Additionally, we have modified the figure size and dashed lines so that the dash lines are adequately distinguishable from the continuous lines.

(28) Equation #7: It is confusing to use P as the number of independent simulations when P is also one of the variables/species in the network. Please consider using different notation.

We sincerely thank the editor(s) and refer(s) for the hhelpful advice!

Now we replace the *P* in current Equation #8 with *Q* and the *P* in current Equation #10 with *W*.

(29) Use of "Detailed Balance": The authors used the term "detailed balance" to describe the intricate balance between the two groups of proteins when forming a polarised pattern. However, "detailed balance" is a term with a specific meaning in thermodynamics. Breaking detailed balance is a feature of nonequilibrium systems, and the polarisation phenomenon is evidently a nonequilibrium process. Using the term "detailed balance" may cause confusion, especially for readers with a physics background. It might be advisable to reconsider the terminology to avoid potential confusion and ensure clarity for readers.

We sincerely thank the editor(s) and referee(s) for the constructive suggestion!

To avoid potential confusion and ensure clarity for readers, now we replace “detailed balance” with “balance”, “required balance”, or “interplay” regarding different contexts.

(30) Terminology: The word "molecule" is used where "molecular species" would be more appropriate, *e.g.*, lines 456 and 551. Please revise these instances.

We sincerely thank the editor(s) and referee(s) for the constructive suggestion!

Now we replace all the “molecule” by “molecular species” as suggested.

(31) Section 2.5: This section is confusing. It isn't clear where the "method outlined" (line 464) is nor what "span an iso-velocity surface at vanishing speed" means in line 470. The sentence in lines 486-488, "An expression similar to Eq. 8 enables quantitative prediction...", is too vague. Please clarify these points and specify what the "similar expression" is and where it can be found.

We sincerely thank the editor(s) and referee(s) for the constructive suggestion!

Now we clarify these points and specify the terms as suggested.

(32) Software Mention: The software is only mentioned in the abstract and conclusions. It should also be mentioned where the computational pipeline is described, and the instructions available in the supplementary information need to be referenced in the main text.

We sincerely thank the editor(s) and referee(s) for pointing out this problem!

Now we mention the software where the computational pipeline is described and reference the instructions available in the Supplemental Text.

(33) Supplementary Material References: Several parts of the supplementary material are never referenced in the main text, including Figure S1, Movies S3-S4, and the Instructions for PolarSim. Please reference these in the main text to clarify their relevance and how they fit with the manuscript's narrative.

We sincerely thank the editor(s) and referee(s) for pointing out this problem!

Now we add all the missing references for supplementary materials to the main text properly.